# Conditional effects of *Cardinium* on microbiota in an invasive whitefly under different ecological factors

Kun Yang,[1,2,3] Cheng-Ran Li,[1,2] Peng-Hao Qin,[1,2] Meng-Ying Yuan,[1,2] Dong Chu[1,2]

**ABSTRACT** *Cardinium*, a vital symbiont in *Bemisia tabaci* Mediterranean (MED), can influence the microbiota in *B. tabaci* MED under high temperatures. However, the effects of *Cardinium* infection on the microbiota in *B. tabaci* MED with different genetic backgrounds under different ecological factors still remain poorly understood. In this study, based on full-length 16S rRNA gene sequencing and quantitative PCR (qPCR) experiments, the effects of *Cardinium* infection on the microbiota in two *B. tabaci* MED geographical populations with different genetic backgrounds were determined with particular attention to ecological factors such as high-temperature treatment and host-plant switching. Results indicated that high temperature treatment and host-plant switching affected the symbiont titer, microbiota diversity, and function differently in the two populations, highlighting the genetic background. The present study also revealed that the increase of *Cardinium* titer would significantly change the response of the microbiota function of the *Cardinium*-infected line compared to the uninfected line, while the decrease and immutability of *Cardinium* titer would not, which indicates that the *Cardinium* titer may be closely associated with the microbiota function in whitefly. Overall, the genetic background of whiteflies influences microbiota response under high temperatures and host-plant changes, and *Cardinium* titer significantly impacts microbiota function. These findings enhance understanding of the complex relationships among symbionts, microbiota, and host insects.

**IMPORTANCE** This study sheds light on how genetic differences in *Bemisia tabaci* Mediterranean (MED) populations influence their microbiota's response to environmental stressors like high temperatures and host-plant changes. By focusing on the role of *Cardinium*—a key symbiont—the research reveals its significant impact on microbiota diversity and function, particularly when its titer increases. The findings emphasize the interplay between genetic background, symbiont levels, and microbiota, advancing our understanding of the ecological adaptability of these insects. This knowledge is vital for developing better pest management strategies and predicting responses to environmental changes.

**KEYWORDS** *Cardinium*, whitefly, genetic background, microbial diversity, high temperature, host-plant switching

**Peer Reviewers** Xiaoli Bing, Nanjing Agricultural University, Nanjing, Jiangsu, China; Yu-Hao Huang, Guangdong Academy of Sciences Institute of Zoology, Guangdong Key Laboratory of Animal Conservation and Resource Utilization, Guangzhou, China

Address correspondence to Dong Chu, chinachudong@qau.edu.cn.

The authors declare no conflict of interest.

See the funding table on p. 17.

*B*emisia tabaci (Gennadius) is a highly invasive and globally distributed insect pest, commonly known as the silverleaf whitefly or sweet potato whitefly. This insect exhibits remarkable biological characteristics, including high fecundity, rapid development, polyphagy, and the ability to develop insecticide resistance quickly, making it a challenging pest to manage (1). *B. tabaci* has at least 40 cryptic species and is notorious for its ability to transmit several plant viruses, including *Begomoviruses*, *Criniviruses*, and *Torradoviruses*, which could cause significant yield losses in various crops worldwide (2–5). *B. tabaci* Middle East-Asia Minor 1 (MEAM1) and Mediterranean (MED) are the two

most invasive and destructive cryptic species all over the world, especially in China (1, 6–9).

Symbionts are widely spread in arthropods, which could have various effects on the biological traits and ecological adaptation of host insects, including fitness alteration and reproductive manipulation (10–13). One key aspect of *B. tabaci* biology that has drawn considerable attention is its intricate relationship with various symbiotic microorganisms. *B. tabaci* can harbor diverse symbionts, including *Portiera*, *Hamiltonella*, *Rickettsia*, *Wolbachia*, *Cardinium*, *Fritschea*, and *Arsenophonus*. These symbionts can influence the biology, ecology, and vector competency for plant viruses of host whiteflies (11, 14–17). As a vital secondary symbiont, *Cardinium* infection significantly changes the host whiteflies' fecundity, fitness, thermotolerance, protein, and miRNA expressions of whiteflies (18–22).

The relationship between symbiotic bacteria and the broader microbiota in insects is complex and dynamic (11, 12, 23, 24). For example, the *Wolbachia* infection would significantly decrease the diversity of microbiota in the host *Sogatella furcifera* and the abundance of many other bacteria (12, 25). *Wolbachia* in *Laodelphax striatellus* would also decrease the bacterial taxon diversity and the abundance of other microbiota (26). Previous research has demonstrated that *Cardinium* infection affects the diversity and function of bacterial communities, such as *Cardinium* infection significantly increasing thermotolerance of host whiteflies (22). Additionally, it can alter the relative abundance of numerous symbionts in whiteflies (27). Understanding the interplay between symbiotic bacteria and other bacterial taxa is essential for unraveling the mechanisms underlying insect-microbe interactions and their ecological and evolutionary implications.

Many biotic and abiotic factors—including diet, host plant, temperature, humidity, and pH—significantly influence the composition, abundance, and activity of symbiotic bacteria in insects (27–32). Among these, temperature is particularly critical, as it profoundly affects microbial diversity and symbiont dynamics (11, 33, 34). For example, the annual mean temperature negatively affects the abundance of many bacteria in *S. furcifera* (12). High temperatures also significantly decrease microbiota diversity and change the symbiont's titers in whiteflies (27). The genetic background of host insects plays a crucial role in shaping their interactions with symbiotic microorganisms (10, 23, 35). Recent research revealed that the host whitefly in China has obvious genetic differentiation (36). The influence of biotic and abiotic factors on microbiota in *B. tabaci* with different genetic backgrounds should be further explored.

Our previous study reported that *Cardinium* infection could confer host *B. tabaci* MED fitness benefits in two geographical populations (Lingshui, LS, and Shouguang, SG populations) (22), with genetic backgrounds distinct from our previous study (36). Yang et al. (27) revealed that *Cardinium* infection can affect the microbiota in the *B. tabaci* MED LS population under high temperature, while the genetic background of the host whitefly and host-plant switching on *B. tabaci* MED microbiota was not fully explored (27). In the current study, based on full-length 16S rRNA gene sequencing and quantitative PCR (qPCR) of two *B. tabaci* MED populations (LS and SG populations), both with one *Cardinium*-infected and one uninfected isofemale line, the influence of host-plant switching and high temperature on the bacterial communities of *B. tabaci* MED was further explored. This research revealed the complex interactions among insects, bacterial communities, temperatures, and host plants, which would enhance the comprehensive understanding of relationships among insects, microbiota, and hosts.

## MATERIALS AND METHODS

### Establishing *Cardinium*-infected and uninfected *Bemisia tabaci* MED populations in the lab

Two *B. tabaci* MED populations were used in this experiment, such as SG populations collected from Shouguang, Shandong Province, China, and the LS population collected from Lingshui, Hainan Province, China. Both populations with one *Cardinium*-infected isofemale line (SGC+ and LSC+) and one uninfected line (SGC− and LSC−) were described in our previous studies (22). As previously reported, SGC+ and LSC+ isofemale lines were detected to be infected with four symbionts (*Portiera*, *Hamiltonella*, *Rickettsia*, and *Cardinium*), while SGC− and LSC− isofemale lines were infected with three symbionts (*Portiera*, *Hamiltonella*, and *Rickettsia*). The cleaved amplified polymorphic sequence of *B. tabaci mtCOI* genes was used to determine the cryptic species of whiteflies (37). Each isofemale line was derived from a single female parent and a male parent (19). Before the experiment, all lines of whiteflies were reared on tobacco plants (*Nicotiana tabacum* L.) (breed: NC89 cultivar) for more than 20 generations. The symbiont infection status was detected in intervals of 30 days and every generation. The rearing condition was 27°C ± 1°C, 16L: 8D, and 60% RH.

### Treatments of *Cardinium*-infected and uninfected *Bemisia tabaci* MED under different plants and temperatures

Though the effects of different temperature on LS population had been reported (27), in this experiment, two isofemale lines of LS or SG population were divided into four treatments (27°C rearing on tobacco, 27°C rearing on cotton, and 31°C rearing on cotton) to determine the influence of temperature and host plants on *Cardinium*-infected and uninfected *B. tabaci*. The experiment was replicated four times. Four tobacco and eight cotton plants (2-true leaf stage) were selected for each replicate experiment. Each plant was cultivated in a 1.5 L plastic pot filled with nutritional soil and isolated in a whitefly-proof screen cage (100 cm × 45 cm × 50 cm). Thirty paired female and male whiteflies from one *B. tabaci* isofemale line were reared on a single host plant and allowed to oviposit for 24 h. After oviposition, all whitefly adults were removed, with the eggs left. After that, four tobacco plants with eggs of four *B. tabaci* isofemale lines were reared at 27°C in an incubator, while half of the eight cotton plants with different whitefly lines were reared in 27°C incubators and half in 31°C incubators. After the eggs emerged, twenty-five 1-day-old whitefly female adults in each plant were randomly selected for 16S rRNA gene sequencing. All treatments were replicated four times simultaneously in the same rearing room (Fig. S1).

### Measurement of different symbiont densities

16S rRNA gene sequencing is widely used for profiling bacterial communities, while its accuracy can be affected by primer bias (some bacterial taxa may be under- or over-represented due to mismatches with universal primers), PCR amplification bias, resolution (closely related species may not be distinguished due to conserved regions in the 16S rRNA gene) (38). Relative qPCR is a sensitive and specific method for quantifying the abundance of target DNA sequences, such as specific bacterial taxa. Its accuracy depends on highly specific primers, accurate PCR efficiency, and results are often normalized to a reference gene (39).

TIANamp Genomic DNA Kit (Tiangen Biotech Co. Ltd., Beijing, China) was used for the DNA extraction of whiteflies. Twenty-five whitefly female adults (1-day-old) in each plant were considered a sample. DNA purities were monitored on 1% agarose gel. All DNA concentrations were adjusted to 10 ng/µL with DEPC water. Four symbiont qPCR primers were used in this experiment (27), and two reference genes of *B. tabaci* MED were used: EF-1α (F: 5′-TAGCCTTGTGCCAATTTCCG-3′, R: 5′-TCCTTCAGCATTACCGTCC3′) (40) and tubulin alpha‑1 chain (F: 5′-ACTGGTGTCCCACTGGGTTC-3′, R: 5′-CGGTGGGT GGTTGGTAGTTG-3′) (41). The geometric average of EF-1α and tubulin alpha‑1 chain

creates an appropriate normalization factor for gene expression experiments in *B. tabaci* MED (42). A qTOWER 2.0/2.2 Real Time PCR Systems (Jena Bioscience GmbH, Thüringen, Germany) with SYBR Premix Ex Taq (Takara Bio Inc., Dalian, China) was used to perform the qPCR reactions. As the symbiont density data all follow the normal distribution, Student's *t*-test (SPSS 21.0) was used to analyze the significant differences in symbiont density between the two treatment groups. The rest of the DNA was prepared for 16S rDNA high-throughput sequencing.

## Preparation and sequencing for 16S rDNA high-throughput gene sequencing

In this experiment, all the operation processes, including DNA amplification, library construction, sequencing, and data analysis, were performed by Biomarker Technologies Corporation, Beijing, China. Briefly, the full length of the bacterial 16S rRNA gene was amplified with the common primer pair (forward primer, 5′-ACTCCTACGGGAGGCAGCA -3′; reverse primer, 5′-GGACTACHVGGGTWTCTAAT-3′) combined with adapter sequences and barcode sequences. PCR amplification was performed in a total volume of 50 µL, which contained 10 µL buffer, 0.2 µL Q5 High-Fidelity DNA Polymerase, 10 µL High GC Enhancer, 1 µL dNTP, 10 µM of each primer, and 60 ng of genome DNA. Thermal cycling conditions were as follows: an initial denaturation at 95°C for 5 min, followed by 15 cycles at 95°C for 1 min, 50°C for 1 min, and 72°C for 1 min, with a final extension at 72°C for 7 min. The PCR products from the first step PCR were purified through VAHTS DNA Clean Beads. A second round of PCR was then performed in a 40 µL reaction, which contained 20 µL 2× Phμsion HF MM, 8 µL ddH$_2$O, 10 µM of each primer, and 10 µL PCR products from the first step. Thermal cycling conditions were as follows: an initial denaturation at 98°C for 30 s, followed by 10 cycles at 98°C for 10 s, 65°C for 30 s, and 72°C for 30 s, with a final extension at 72°C for 5 min. Finally, all PCR products were quantified by Quant-iT dsDNA HS Reagent and pooled together. The amplified library was sequenced using a PacBio SMRT RS II DNA sequencing platform (Pacific Biosciences, Menlo Park, CA, USA).

The raw reads generated from sequencing were filtered and demultiplexed using the SMRT Link software (version 8.0) with the minPasses ≥5 and minPredictedAccuracy ≥0.9, in order to obtain the circular consensus sequencing (CCS) reads. Subsequently, the lima (version 1.7.0) was employed to assign the CCS sequences to the corresponding samples based on their barcodes. CCS reads containing no primers and those reads beyond the length range (1,200–1,650 bp) were discarded through the recognition of forward and reverse primers and quality filtering using the Cutadapt quality control process (version 2.7).

## Data analysis for 16S rDNA high-throughput gene sequencing

Low-quality or off-target sequences were filtered by PacBio circular consensus sequencing technology (43). The phylogenetic tree and the diversity indices, including alpha diversity indices (Ace, Chao1, Shannon, and Simpson) and beta diversity (principal coordinate analysis [PCoA] and heatmaps based on Bray-Curtis similarity analysis) were all analyzed by USEARCH (v10.0) (44), and QIIME software (45) at BMK Cloud (https://www.biocloud.net/). The phylogenetic analysis involved selecting representative sequences from the highest abundance operational taxonomic units at different taxonomic levels using QIIME results, performing multiple sequence alignment, and constructing phylogenetic trees where branch lengths represent evolutionary distances between species. Visualization was done using matplotlib 1.4.3 (https://matplotlib.org/1.4.3/). PICRUSt2 (Phylogenetic Investigation of Communities by Reconstruction of Unobserved States) was used to predict functions of bacteria based on full-length 16S rRNA genes in the Kyoto Encyclopedia of Genes and Genomes database (https://www.kegg.jp/) (46). The Shapiro-Wilk test (SPSS 21.0) was performed to measure whether the alpha diversity index data followed the normal distribution. As the alpha diversity index data all follow the normal distribution, Student's *t*-test (SPSS 21.0) was used to analyze the significant differences in the alpha index between the two treatment groups.

## RESULTS

### Composition of bacterial communities in *Bemisia tabaci*

Based on full-length 16S rRNA gene sequencing results of LS and SG *B. tabaci* populations under different treatments, the Raw CCS of all samples was higher than 8,042 sequences. The Effective CCS of all samples was higher than 7,282 sequences, and the effective rates (the percentage of Effective CCS in Raw CCS) were above 88%. The average length of sequenced CCS in each sample ranged from 1,425 to 1,466 bp (Table S1). A total of 35 genera and 45 species of bacteria were observed in all samples. Four symbionts were detected, including one primary symbiont, *Candidatus* Portiera aleyrodidarum, and three secondary symbionts (*Candidatus* Cardinium hertigii, *Rickettsia* sp. strain MEAM1, and *Candidatus* Hamiltonella defensa) (Fig. 1). Each symbiont genus detected in this study contained only a single symbiont species; we did not detect multiple symbiont species within any single genus, nor did we observe any cases of superinfection (i.e., simultaneous infection by multiple symbiont species of the same genus) in the samples. Most bacterial genera were related to the phylum Proteobacteria, Bacteroidetes, and Firmicutes. As for symbionts, *Portiera aleyrodidarum*, *Hamiltonella defensa,* and *Rickettsia* sp. strain MEAM1 were all related to the phylum Proteobacteria, while *Cardinium hertigii* was associated with Bacteroidetes. The *Portiera* and *Hamiltonella* were closely related based on full-length 16S rRNA genes (Fig. 1).

With PCoA, all replicates of the same treatment were clustered together, whether in *Cardinium*-infected or uninfected whitefly isofemale lines in LS and SG populations (Fig. S2). The LS population (Fig. S2A and B) and SG population (Fig. S2C and D) have four treatments with different temperatures and host plants. Based on the PCoA results, it was observed that although the four replicates of different treatments within the same *B. tabaci* isofemale lines could not be distinguished from each other, they do exhibit distinct characteristics (Fig. S2).

As for the composition of bacterial communities in whiteflies, most bacteria were related to the phylum Proteobacteria at the phylum level (as most symbionts were related to the phylum Proteobacteria) (Fig. S3). Regarding the composition of bacterial communities at the species level, symbionts tend to make up the majority (more than 92%) of the bacterial communities found in all whitefly samples (Fig. 2). *Portiera aleyrodidarum* and *Hamiltonella defensa* were infected in all whitefly samples, while *Cardinium hertigii* was only infected in LSC+ and SGC+ isofemale lines. On the other hand, *Rickettsia* sp. strain MEAM1 was present in a large proportion in all samples, except for the SGC+ line at 27°C on tobacco. In this treatment, the relative abundance of *Rickettsia* sp. strain MEAM1 was sharply reduced. However, the relative abundance of *Rickettsia* did not sharply decrease in the SGC− line and the two LS population lines under the same rearing conditions (Fig. 2). PCR detection using symbiont-specific primers on DNA extracted from host plant leaves (cotton and tobacco) did not reveal the presence of any symbionts, indicating that these bacteria are not found in the host plants.

### Effects of high temperature on symbionts' titers and diversity of *Bemisia tabaci* microbiota

To determine which factors influence the titer of symbionts in *B. tabaci*, this study conducted a qPCR experiment on four symbionts (*Portiera aleyrodidarum*, *Hamiltonella defensa*, *Cardinium hertigii*, and *Rickettsia* sp. strain MEAM1) of *B. tabaci* with different genotypes under different temperature conditions. As shown in Fig. 3, the high temperature significantly changes the titer of symbionts. The effects of high temperatures on symbionts in LS whiteflies were previously reported (27). Based on the qPCR results, the high temperature significantly increases the titer of *Cardinium hertigii* but decreases the titer of *Rickettsia* sp. strain MEAM1 in the LSC+ line, while no significant change in symbiont titer is observed in the LSC− line (Fig. 4C and D). While for the SG population, *Portiera aleyrodidarum* titer in SGC− line under 31°C in cotton (9.95 ± 0.26, Mean ± SEM) was significantly higher than that under 27°C in cotton (4.73 ± 0.70, Mean ± SEM) ($t_{6/0.05}$

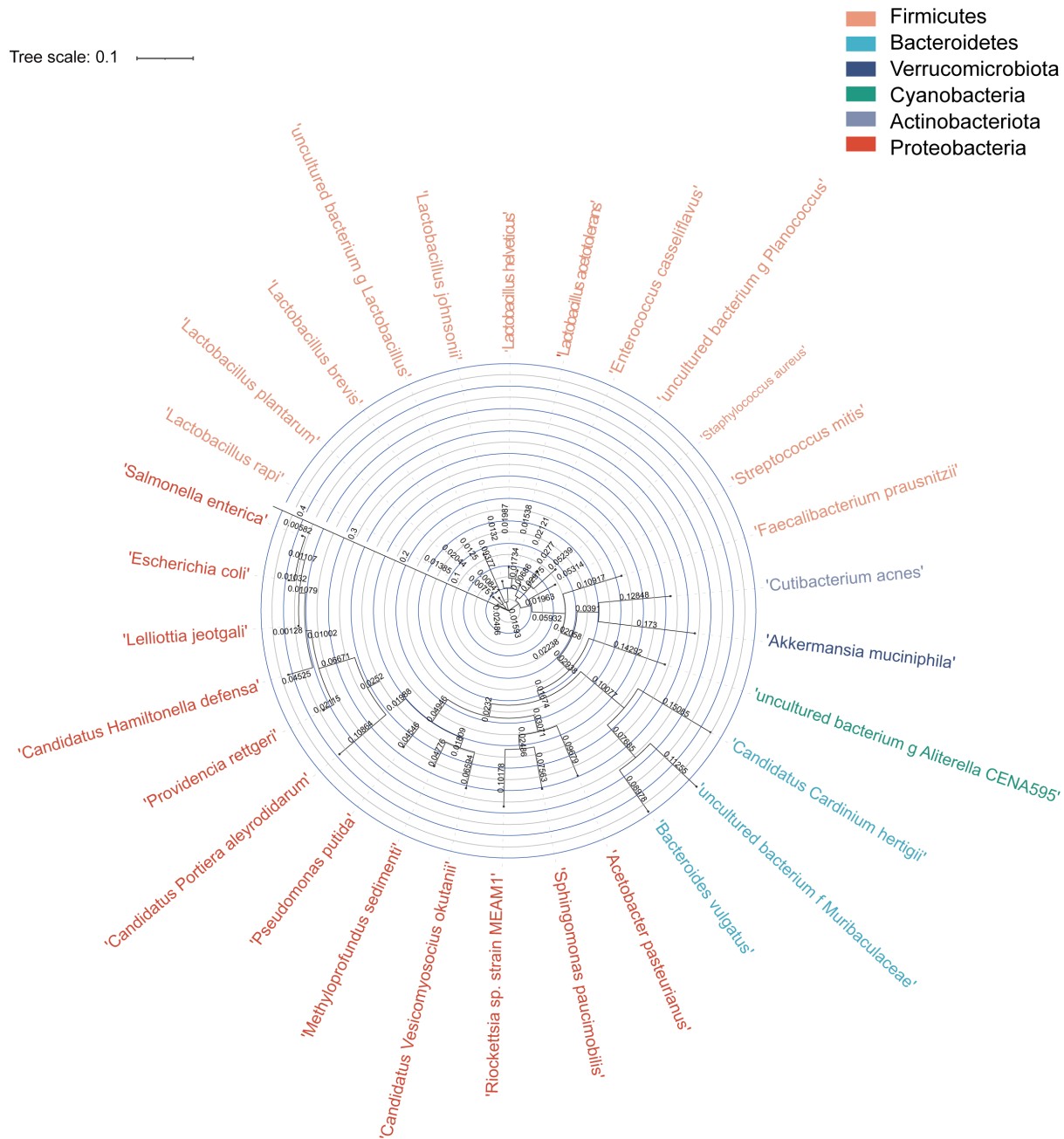

**FIG 1** Circular phylogenetic tree showing the evolutionary relationships among bacterial species isolated from *Bemisia tabaci* MED individuals from the LS and SG populations. The tree was constructed using QIIME (v1.9.0) based on 16S rRNA gene sequencing data. Each tip of the tree represents a bacterial genus or species, with branch colors corresponding to different bacterial phyla as indicated in the legend. Closely related taxa are clustered together, illustrating their shared evolutionary history. Branch lengths reflect evolutionary distances between taxa.

= 6.071, *P* < 0.01, Student's *t*-test) (Fig. 3A). The titer of *Hamiltonella defensa* in 31°C in cotton of SGC- line (16.17 ± 4.59, Mean ± SEM) was markedly higher than that under 27°C in cotton (1.45 ± 0.30, Mean ± SEM) (t$_{6/0.05}$ = 3.816, *P* < 0.05, Student's *t*-test) (Fig. 3B). Similarly, the *Rickettsia* sp. strain MEAM1 titer in SGC− line under 31°C in cotton (31.52 ± 5.57, Mean ± SEM) was substantially increased compared to that under 27°C in cotton (6.31 ± 0.76, Mean ± SEM) (t$_{6/0.05}$ = 5.310, *P* < 0.01, Student's *t*-test) (Fig. 3D). No significant influence of high temperature on symbionts in SGC+ line detected (Fig. 3). The effects of high temperature on symbiont tiers also changed with host whitefly genotype and *Cardinium*-infected status.

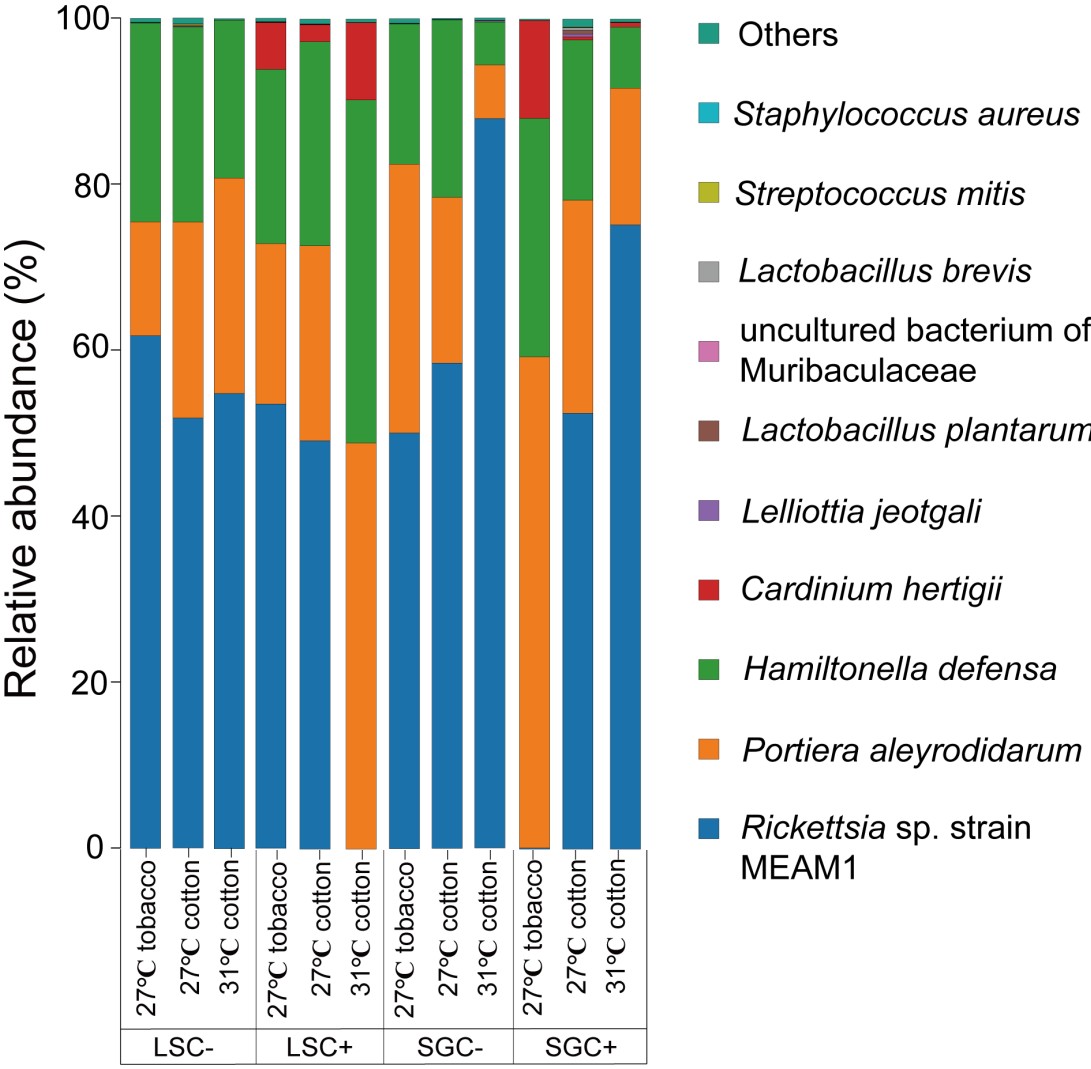

**FIG 2** Relative abundance of top 10 bacterial species in LS and SG population *Bemisia tabaci* MED under different treatment conditions, detected by full-length 16S rRNA gene sequencing.

Four alpha diversity indexes (Shannon, Simpson, Chao1, and ACE indexes) of microbiota in *B. tabaci* were compared in *Cardinium*-infected and uninfected LS and SG populations under different temperature conditions (Fig. 4). The effects of high temperature on microbial diversities of LS population whiteflies were previously reported (27). The Shannon index of SGC− line under 31°C in cotton (0.66 ± 0.13, Mean ± SEM) was also significantly lower than that under 27°C in cotton (1.41 ± 0.05, Mean ± SEM) ($t_{6/0.05}$ = −5.328, $P < 0.01$, Student's *t*-test); similarly, SGC+ line under 31°C in cotton (1.07 ± 0.04, Mean ± SEM) was significantly lower than that under 27°C in cotton (1.60 ± 0.08, Mean ± SEM) ($t_{6/0.05}$ = −5.690, $P < 0.01$, Student's *t*-test). (Fig. 4A). For the Simpson diversity index, the index of SGC− line under 31°C (0.21 ± 0.05, Mean ± SEM) was also significantly lower than that under 27°C in cotton (0.57 ± 0.03, Mean ± SEM) ($t_{6/0.05}$ = −6.530, $P < 0.01$, Student's *t*-test), and the index of SGC+ line under 31°C in cotton (0.39 ± 0.03, Mean ± SEM) was substantially reduced compared to that under 27°C in cotton (0.61 ± 0.01, Mean ± SEM) ($t_{6/0.05}$ = −7.449, $P < 0.001$, Student's *t*-test) (Fig. 4B). The ACE index shows no significant differences whether in LS or SG populations (Fig. 4C). Chao1 index of SGC− line under 31°C in cotton (12.16 ± 3.81, Mean ± SEM) was significantly lower than that under 27°C in cotton (27.44 ± 4.08, Mean ± SEM) ($t_{6/0.05}$ = −2.448, $P < 0.05$, Student's *t*-test) (Fig. 4D). In total, high temperature would significantly decrease the diversity of

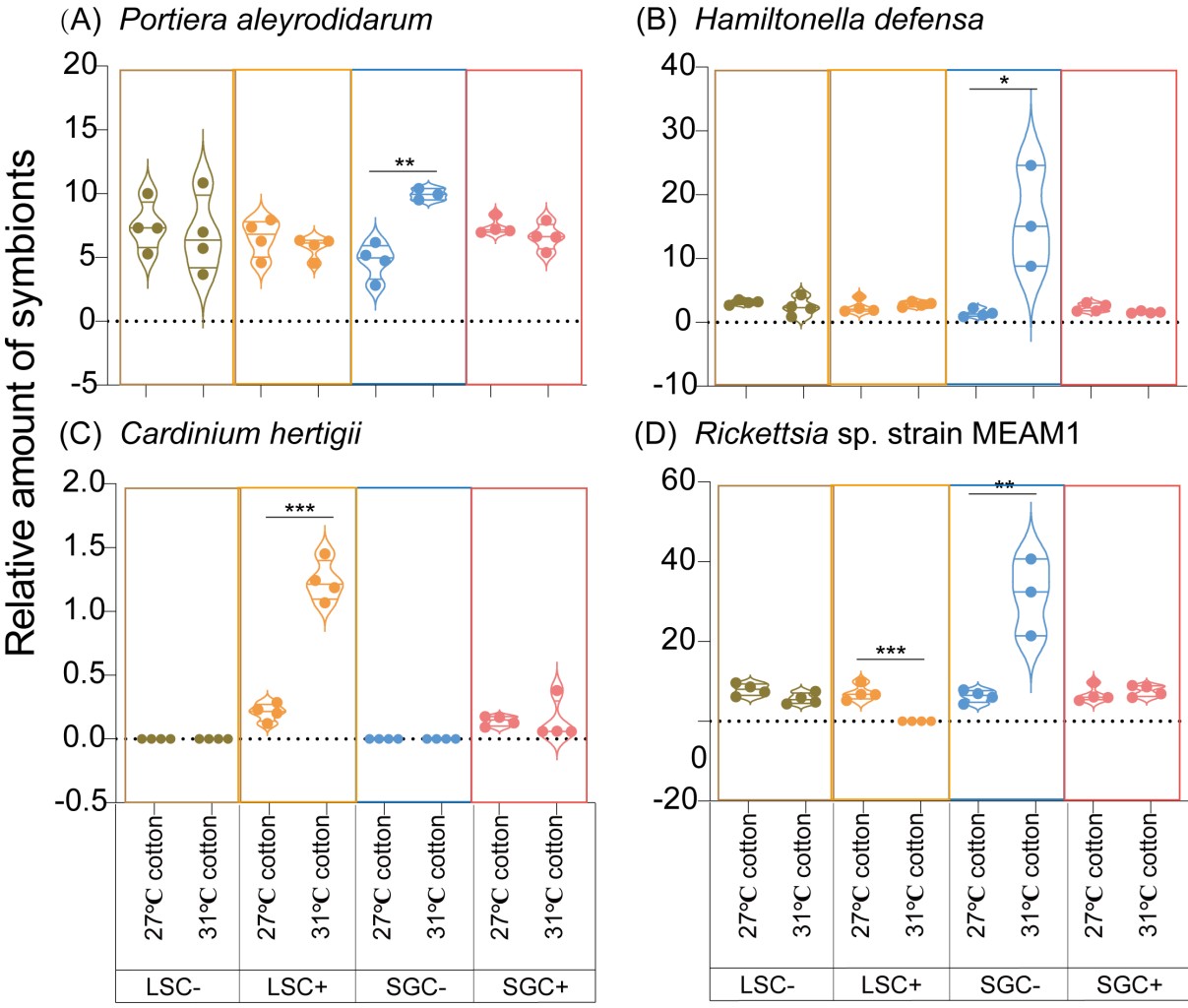

**FIG 3** Relative levels of symbiont density in LS and SG populations of *Bemisia tabaci* MED grown under different temperatures. Four symbionts are shown: *Portiera aleyrodidarum* (A), *Hamiltonella defensa* (B), *Cardinium hertigii* (C), and *Rickettsia* sp. strain MEAM1 (D).

microbiota in *B. tabaci*, which effects could alter with different *B. tabaci* genotypes and *Cardinium*-infected status.

Cardinium had different effects on *B. tabaci* with different genetic backgrounds under high temperature and host-plant switching. In all *B. tabaci* MED samples, most genes of bacterial communities were related to metabolic function. The other genes were mainly associated with functions of Genetic Information Processing and Environmental Information Processing, and the functions were varied in different treatments (Fig. S4).

As shown in Fig. 5, in the LSC– line, no significant effects of high temperature on functions of microbiota in *B. tabaci* MED were detected (Fig. 5A), while in the SGC– line, high temperature (31°C) would also significantly decrease the genes related to Environmental Information Processing, Cellular Processes, and Metabolism but increase substantially the genes related to Organismal Systems and Genetic Information Processing of SGC– line whiteflies (Fig. 5B). For the LSC+ line, high temperature would significantly increase the proportion of microbiota genes related to Metabolism but significantly decrease the gene proportion related to Organismal Systems and Genetic Information Processing (Fig. 5C). In the SGC+ isofemale line, *B. tabaci* reared at 31°C on cotton has significantly larger proportions of genes related to Organismal Systems and Genetic Information Processing, but fewer genes associated with Environmental Information Processing and Metabolism compared to *B. tabaci* reared in 27°C on cotton (Fig. 5D).

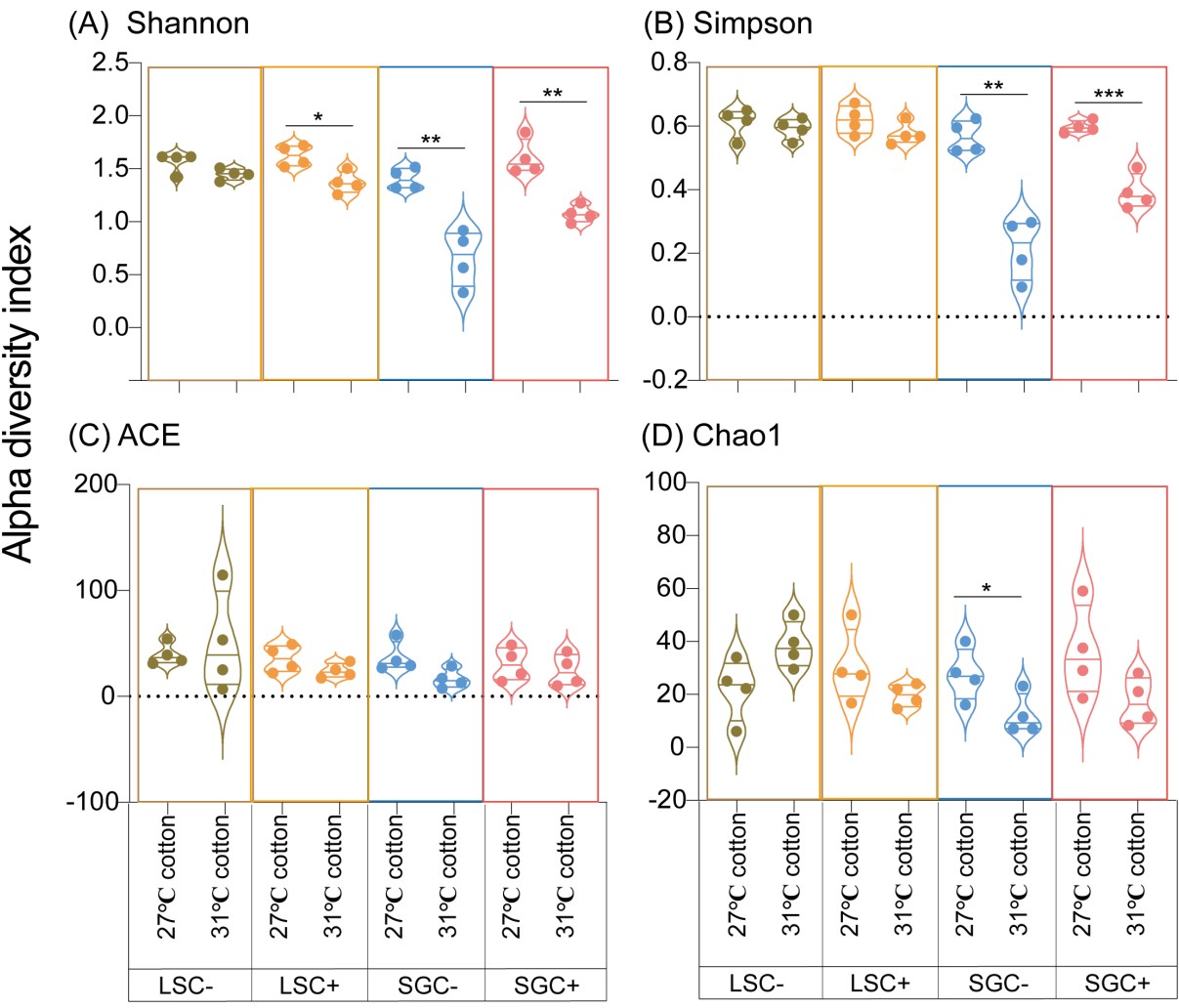

**FIG 4** Alpha diversity index of bacterial communities in LS and SG populations of *Bemisia tabaci* MED grown under different temperatures. The four diversity indices include Shannon (A), Simpson (B), ACE (C), and Chao1 (D).

## Effects of host-plant switching on symbionts' titers and diversity of *Bemisia tabaci* microbiota

The effects of host-plant switching on symbiont titer were clarified with *B. tabaci* of different genotypes and *Cardinium*-infected status (Fig. 6). Based on the results, *Portiera aleyrodidarum* titer in LSC− line on tobacco (3.37 ± 0.20, Mean ± SEM) was significantly higher than that on cotton (7.48 ± 0.97, Mean ± SEM) ($t_{6/0.05}$ = −4.149, *P* < 0.01, Student's *t*-test), while *Portiera aleyrodidarum* titer in SGC− line on tobacco (11.53 ± 1.01, Mean ± SEM) was significantly increased compared to that on cotton (4.73 ± 0.70, Mean ± SEM) ($t_{6/0.05}$ = 5.513, *P* < 0.01, Student's *t*-test). Interestingly, the titer of *Portiera aleyrodidarum* would not change in *Cardinium*-infected lines (LSC+ and SGC+) (Fig. 6A). These results showed a significant influence of *Cardinium* infection on *Portiera aleyrodidarum* titer, which also implied a vital role of host whitefly genotype on the titer of *Portiera aleyrodidarum*.

The titer of *Hamiltonella defensa* on tobacco of LSC− line (1.75 ± 0.13, Mean ± SEM) was substantially higher than that on cotton (3.15 ± 0.18, Mean ± SEM) ($t_{6/0.05}$ = −6.301, *P* < 0.001, Student's *t*-test). Similarly, the titer of *Hamiltonella defensa* on tobacco of SGC+ line (27.25 ± 3.39, Mean ± SEM) was markedly increased compared to that on cotton (2.94 ± 0.49, Mean ± SEM) ($t_{6/0.05}$ = 7.313, *P* < 0.001, Student's *t*-test) (Fig. 6B). The *Cardinium hertigii* titer of the LSC+ line under tobacco (1.00 ± 0.11, Mean ± SEM) was

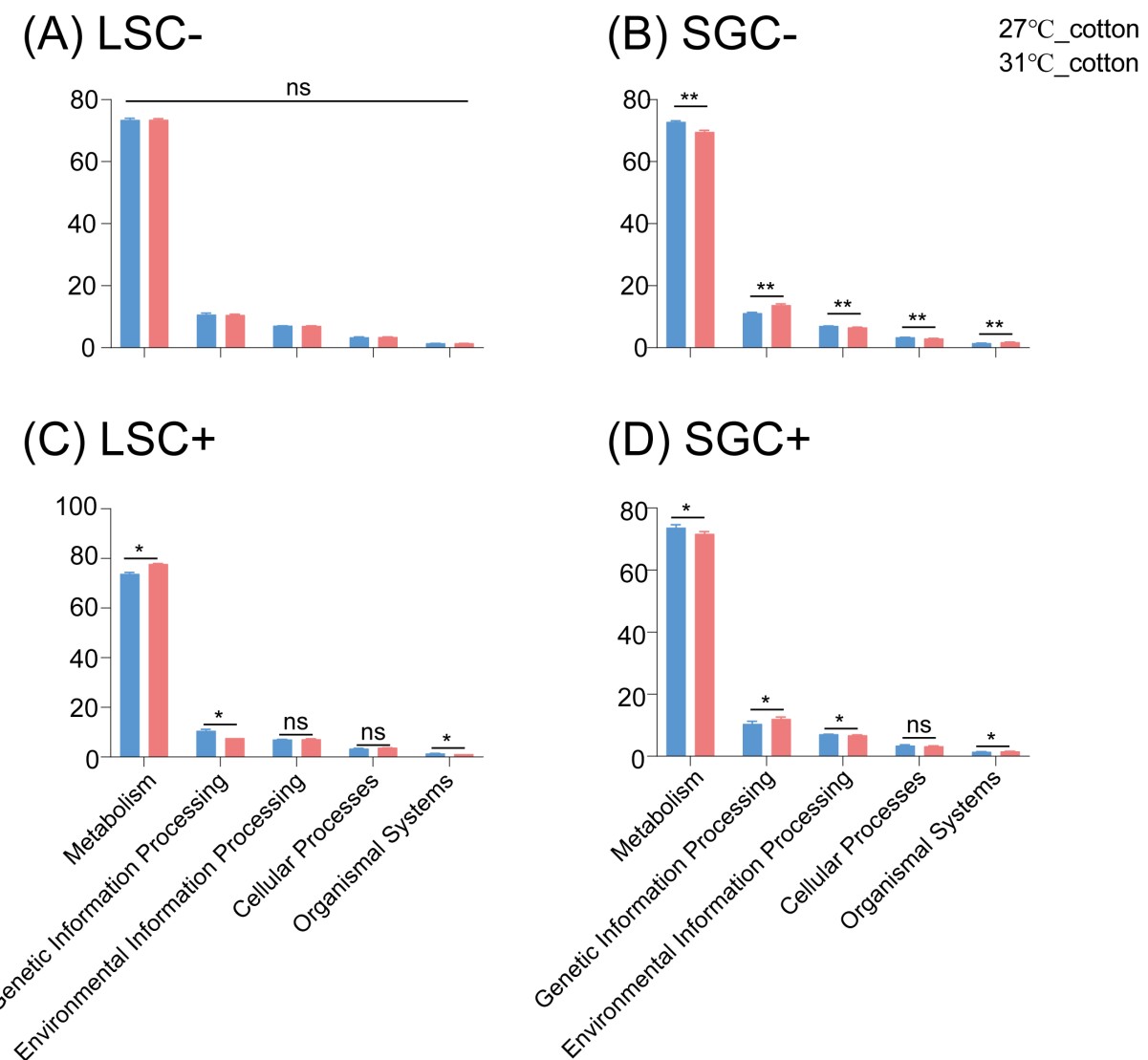

**FIG 5** Significant differences in the functional analysis results for bacterial communities in LS and SG populations, *Cardinium*-infected and uninfected isofemale lines *Bemisia tabaci* MED under different temperatures. The four *Bemisia tabaci* MED lines include LSC− (A), SGC− (B), LSC+ (C), and SGC+ (D) are shown, respectively.

substantially higher than that on cotton (0.21 ± 0.03, Mean ± SEM) ($t_{6/0.05}$ = 6.863, $P <$ 0.001, Student's *t*-test), while the *Cardinium hertigii* titer of the SGC+ line under tobacco (0.01 ± 0.005, Mean ± SEM) was significantly lower than that on cotton (0.14 ± 0.02, Mean ± SEM) ($t_{6/0.05}$ = −6.197, $P <$ 0.001, Student's *t*-test) (Fig. 6). *Rickettsia* sp. strain MEAM1 titer in SGC+ line under tobacco (0.002 ± 0.002, Mean ± SEM) was markedly lower than that on cotton (6.77 ± 1.01, Mean ± SEM) ($t_{6/0.05}$ = −6.685, $P <$ 0.001, Student's *t*-test) while *Rickettsia* sp. strain MEAM1 titer in SGC− line under tobacco (11.69 ± 1.51, Mean ± SEM) is significantly higher than that on cotton (6.31 ± 0.76, Mean ± SEM) ($t_{6/0.05}$ = 3.179, $P <$ 0.05, Student's *t*-test) (Fig. 6D). The qPCR results with host-plant switching suggested the importance of host *B. tabaci* genetic backgrounds and *Cardinium* infection on symbiont titers in *B. tabaci*.

Diversity of microbiota in *B. tabaci* was compared in *Cardinium*-infected and uninfected LS and SG populations reared on cotton and tobacco plants (Fig. 7). Based on the results, Shannon index in LSC− line on tobacco (1.37 ± 0.02, Mean ± SEM) was significantly lower than that on cotton (1.56 ± 0.05, Mean ± SEM) ($t_{6/0.05}$ = 3.872, $P <$ 0.01, Student's *t*-test) under 27℃, while the Shannon index of SGC+ line on tobacco (1.35 ±

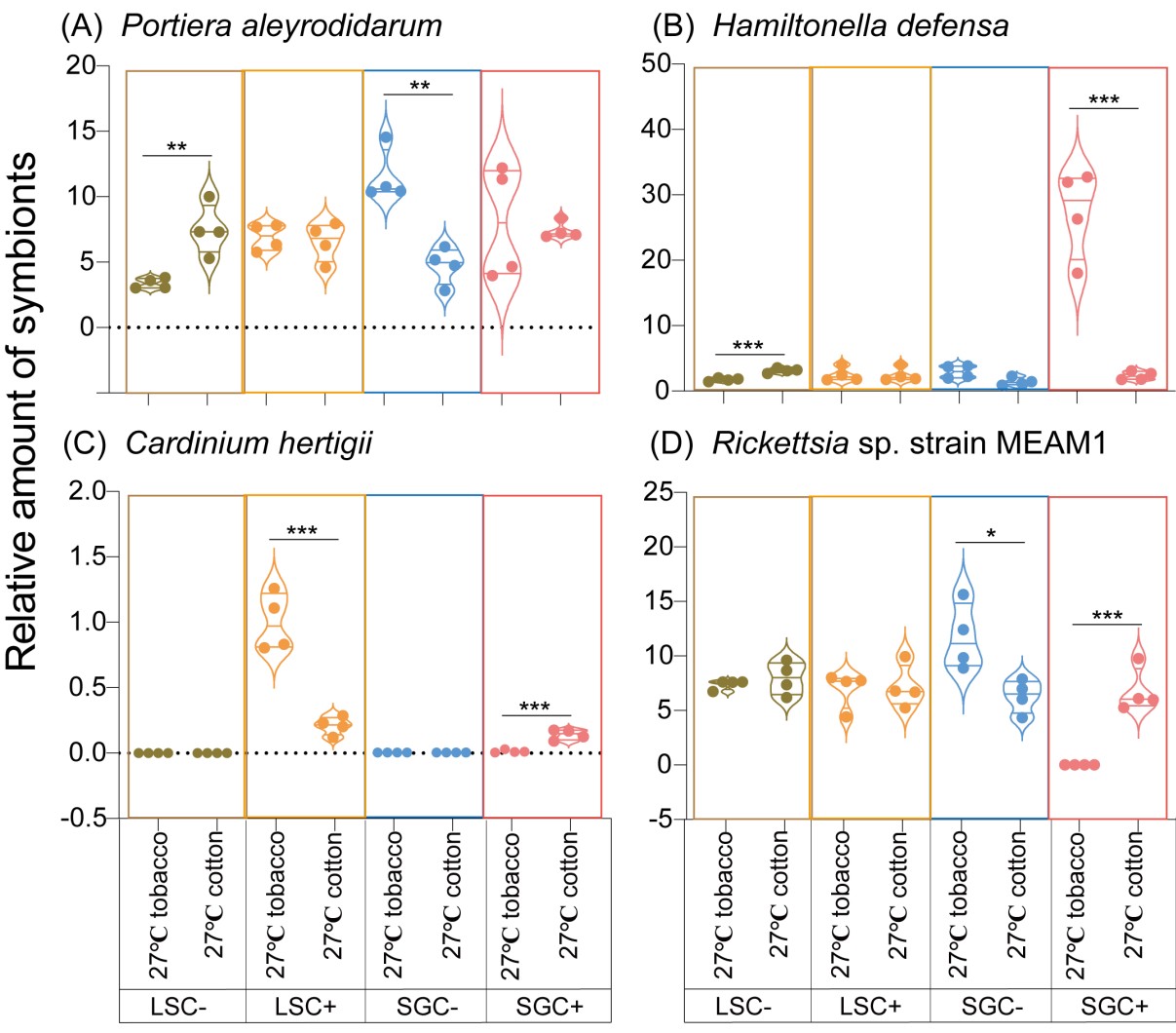

**FIG 6** Relative levels of symbiont density in LS and SG populations of *Bemisia tabaci* MED grown under different host plants. Four symbionts are shown: *Portiera aleyrodidarum* (A), *Hamiltonella defensa* (B), *Cardinium hertigii* (C), and *Rickettsia* sp. strain MEAM1 (D).

0.04, Mean ± SEM) was also significantly lower than that on cotton (1.60 ± 0.08, Mean ± SEM) ($t_{6/0.05}$ = 2.710, $P < 0.05$, Student's *t*-test). For LSC+ and SGC– lines, no significant differences were detected (Fig. 7A). Simpson index of the LSC– line under tobacco (0.54 ± 0.01, Mean ± SEM) was significantly lower than that on cotton (0.61 ± 0.02, Mean ± SEM) ($t_{6/0.05}$ = 2.974, $P < 0.05$, Student's *t*-test). Similarly, the Simpson index of the SGC+ line on tobacco (0.55 ± 0.02, Mean ± SEM) was also dramatically decreased compared to that on cotton (0.61 ± 0.01, Mean ± SEM) ($t_{6/0.05}$ = 2.671, $P < 0.05$, Student's *t*-test). LSC– line under tobacco (0.54 ± 0.01, Mean ± SEM) was significantly lower than that on cotton (0.61 ± 0.02, Mean ± SEM) ($t_{6/0.05}$ = 2.974, $P < 0.05$, Student's *t*-test), and in LSC+ and SGC– lines, no significant differences were detected (Fig. 6B). As for ACE and Chao1 indexes, no significant differences were detected between whiteflies on different plants whether in LS or SG populations (Fig. 6C and D). In total, host-plant switching from tobacco to cotton would significantly increase the diversity of microbiota in whitefly, whose influence was also related to *B. tabaci* genetic backgrounds and *Cardinium*-infected status.

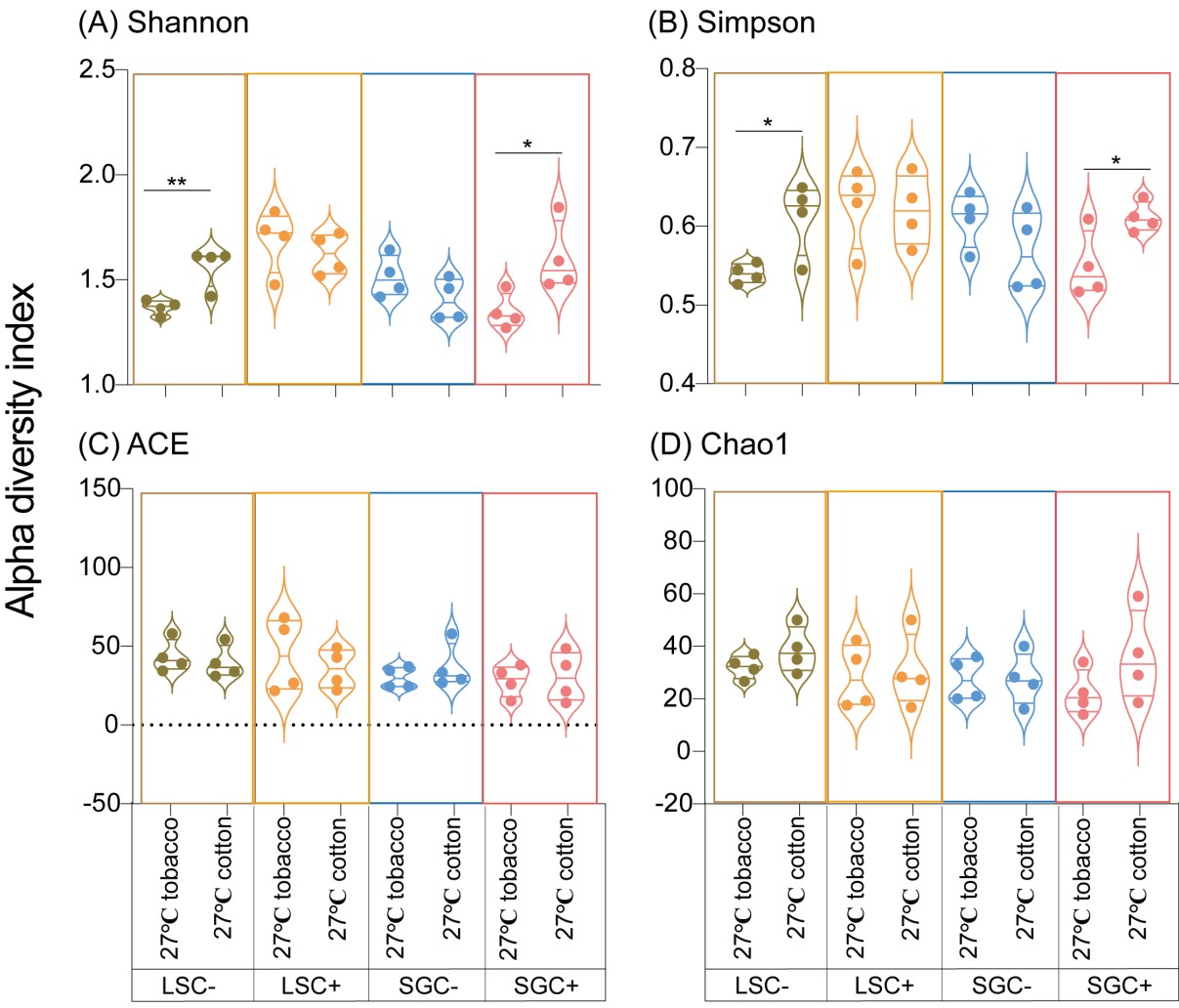

**FIG 7** Alpha diversity index of bacterial communities in LS and SG populations of *Bemisia tabaci* MED grown under different host plants. The four diversity indices include Shannon (A), Simpson (B), ACE (C), and Chao1 (D).

## Effects of host-plant switching on microbiota functions in *Cardinium*-infected and uninfected *Bemisia tabaci*

The effects of host-plant switching on microbiota function in *B. tabaci* MED were also explored (Fig. 8). Based on the results, in two *Cardinium*-uninfected lines (LSC− and SGC−), host-plant switching from tobacco to cotton would not significantly change the function of the microbiota of whiteflies (Fig. 8A and B). As for two *Cardinium*-infected lines, host-plant switching would not alter the microbiota function of the LSC− line, either (Fig. 8C), while the SGC+ isofemale line *B. tabaci* reared on tobacco has significantly lower proportions of genes related to Genetic Information Processing and Organismal Systems compared to *B. tabaci* reared on cotton under 27°C, while *B. tabaci* reared on tobacco has a significantly higher proportion of Metabolism genes than cotton ones (Fig. 8D). All these results suggested a vital influence of host genetic backgrounds and *Cardinium*-infected status on the microbiota function of whiteflies under different temperatures.

## DISCUSSION

In this experiment, we used full-length 16S rRNA gene sequencing and qPCR to detect the bacterial communities and relative amounts of symbionts in *Cardinium*-infected

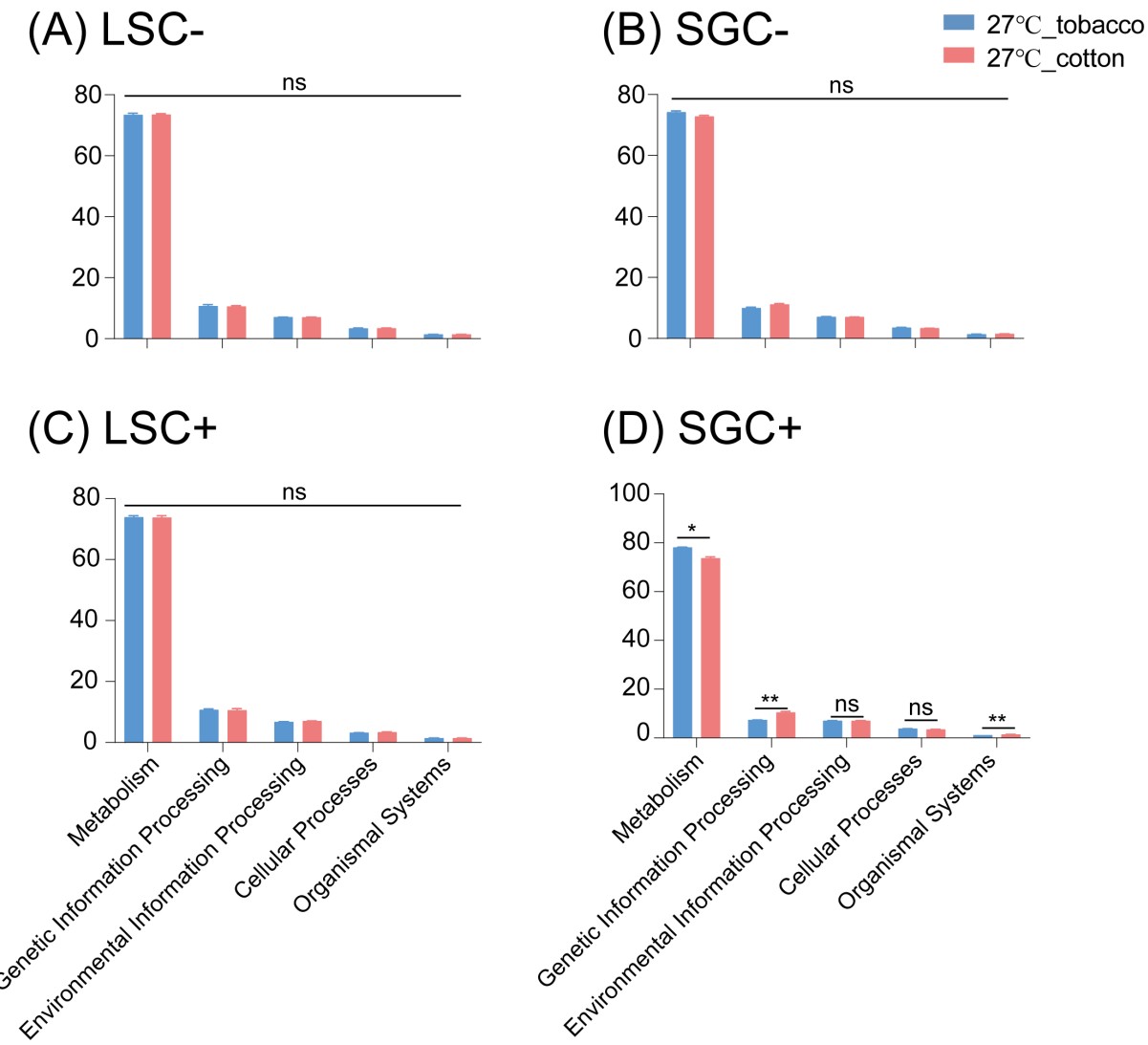

**FIG 8** Significant differences in the functional analysis results for bacterial communities in LS and SG populations, *Cardinium*-infected and uninfected isofemale lines under different host plants. The four *Bemisia tabaci* MED lines include LSC– (A), SGC– (B), LSC+ (C), and SGC+ (D) are shown, respectively.

and uninfected *B. tabaci* populations from LS and SG under different host plants and temperatures. The results showed that high temperature significantly reduced the diversity of bacterial communities and significantly altered the abundance of symbionts in the *Cardinium*-infected LS and SG whitefly populations, regardless of *Cardinium* infection status and isofemale lines. Moreover, the host-plant switching of *B. tabaci* from tobacco to cotton significantly affected the relative abundance of symbionts in both SG and LS populations. Conversely, host-plant switching and high temperature influenced bacterial communities' functions in the whiteflies of SGC+ and SGC– isofemale lines.

## Composition of microbiota in *Bemisia tabaci*

In our previous study, we found that the bacterial phylum Proteobacteria was highly prevalent in *B. tabaci* MED, accounting for over 90% of the total bacterial population. We also observed a substantial abundance of the genus *Rickettsia* (27, 47). These results were consistent with our current findings. Meanwhile, except for the primary symbiont *Portiera*, the secondary symbiont *Hamiltonella* was abundant in all whitefly samples. This phenomenon is not unexpected, as *Hamiltonella* has been observed to play a vital role in whiteflies. *Hamiltonella* can not only provide host whiteflies with essential vitamins or

cofactors, but also compensate for the missing steps in many pathways of the primary symbiont, *Portiera* (16, 48). These results showed that the secondary symbiont *Hamiltonella* was important for the growth and development of *B. tabaci* MED.

## Effects of high temperature on microbiota in *Bemisia tabaci*

Ma et al. (49) reported that extremely high temperatures (EHTs) frequently occurred worldwide. Therefore, further research is needed to understand the effects of EHTs on microbiota, particularly symbionts in insects (49). The high temperature significantly influences the titer of symbionts in insect hosts (24, 50, 51). In this experiment, *Rickettsia* titer was significantly decreased under 31℃ compared to 27℃ of the LSC+ line (Fig. 3D). Our results are similar to those in aphids (52) and mosquitoes (53), where symbiont titers were also decreased under heat stress. However, the titer of *Cardinium* of the LSC+ line, *Hamiltonella* titer, and *Rickettsia* titer of the SGC− line significantly increased on 31℃ cotton (Fig. 4). This is not surprising, as high temperatures can stimulate the metabolic activity of both the insect host and symbiotic microorganisms. This heightened metabolic rate may accelerate symbionts' growth and reproduction, increasing their titer within the host (54, 55).

In this study, the diversity of bacterial communities significantly decreased under high temperatures (Fig. 4). These results are consistent with our previous findings that the microbiota diversities sharply reduced in the LS population (27). Temperatures, especially EHTs, always significantly alter the structures of microbiota in insects (11, 33, 34). For example, in Eastern subterranean termites, the microbial diversity and bacterial abundance significantly decrease with increasing temperature (56). High temperature negatively affects the microbial diversity of the white-backed planthopper, *Sogatella furcifera* (12). The intricate impact of temperature on the microbiota in insects requires further investigation.

## Effects of host-plant switching on microbiota in *Bemisia tabaci*

Host plants are important for phytophagous insects. There is a strong interaction between insects and host plants, especially sap-feeding insects, which could feed on the host plants' vascular system (57–59). Different host plants influenced the diversity of the bacterial community as well as the densities and infections of symbionts in spider mites (60, 61). As for *Bemisia tabaci*, previous research revealed that host plants affect the diversity of bacterial communities (47). The relative amounts of symbionts *Portiera*, *Hamiltonella*, and *Rickettsia* varied in different host plants (31, 32, 62). In this experiment, the microbiota diversities of LSC− and SGC+ lines of whiteflies were significantly influenced by different host plants (Fig. 6), while the symbiont densities also changed with the host-plant switching (Fig. 5). Liu et al. (62) suggested that nitrogen nutrition of host plants could affect the abundance of symbionts, especially for *Portiera* (62). The density of *Rickettsia* in whiteflies could be influenced by the expression of vitellogenin (63). Additionally, another research study suggested that gossypol, a secondary metabolite of cotton, could suppress symbiont density in aphids (64). The variations in the symbionts could be complex due to the combined effects of various factors in this experiment.

## Effects of *Cardinium* infection on microbiota in *Bemisia tabaci*

The effects of host-plant switching and high temperature on *Cardinium*-infected and uninfected *B. tabaci* were not always congruent. For example, the *Rickettsia* abundance sharply decreased after the plant switch of the SGC− line while significantly increasing the SGC+ line under the same conditions (Fig. 5D), which indicates that *Cardinium* has a vital effect on the titers of other symbionts.

The relationships between different symbionts within arthropod hosts are complex. For example, co-infection of *Wolbachia* and *Spiroplasma* in fields was significantly higher than expected crossed populations in *Tetranychus truncatus* (61). The *Hamiltonella* and

*Cardinium* exhibit an antagonistic interaction in whiteflies (65). *Wolbachia* negatively influences many bacteria abundance in *S. furcifera* (12) and *Laodelphax striatellus* (26). Within the same host insects, symbionts, such as bacteriocytes, consistently reside in the same area. This can lead to competition for limited space and nutrition (23, 24), resulting in a competitive relationship between different symbionts within the same host.

Microbial communities, especially for symbionts, are important for the fitness, reproductive manipulations, resistance to adverse factors, metabolism, and many other aspects of host insects (11, 22, 23, 66). In *B. tabaci*, *Cardinium* infection significantly increases the thermotolerance of hosts by up-regulating thermotolerance-related genes (22). The *Hamiltonella* could alter the expression level of histone H3 lysine 9 trimethylation to influence the sex ratio of host *B. tabaci* (67). Primary symbiont *Portiera* controls the level and localization of *Vitellogenin* in host *B. tabaci* (68). In this study, the host-plant switching and high temperature significantly changed the functional genes of host whiteflies of the SGC+ and SGC− lines (Fig. 5 and 8). These results are consistent with the previous study, as the functional genes of the LSC+ line significantly changed after heat stress treatment (27). However, the effects of *Cardinium* on whitefly functions should be further clarified.

Symbiont titer often correlates with the functional roles performed by symbionts in insects, such as nutrient provisioning and protection against pathogens. Higher symbiont titers always enhance the efficiency of the functions (69–72). For example, in aphids, the titer of the symbiont *Buchnera* is crucial for providing essential amino acids or vitamins to the host (69). Manipulation of symbiont titer through antibiotic treatment or genetic engineering can elucidate the direct impacts on symbiotic functions (72). In a previous study, we determined that in both LS and SG populations, *Cardinium* infection could protect host *B. tabaci* from heat stress, while *Cardinium* could provide stronger protection in the LS population compared to that in the SG population (22). Here, based on qPCR results, *Cardinium* titer in the LSC+ line significantly increased in high temperature condition, while *Cardinium* titer did not increase in the SGC+ line (Fig. 3C), and the response of microbiota function to high temperature of the LSC+ line is different from the LSC− line (Fig. 5A and C), while the response of microbiota function in the SGC+ line is similar to the SGC− line after high temperature (Fig. 5B and D). These results suggest a correlation between *Cardinium* titer and functions of the microbiota, especially for the thermotolerance provided by *Cardinium* in *B. tabaci*. As for the host-plant switching experiment, an increase in *Cardinium* titer in the SG population could alter the response of the SGC+ line compared to the SGC− line. In contrast, a decrease in *Cardinium* titer did not change the response of the LSC+ line compared to the LSC− line (Fig. 6 and 8). These results further confirm the vital influence of *Cardinium* titer on microbiota function.

## Effects of host genetic background on microbiota in *Bemisia tabaci*

Another noticeable issue is variations in host-plant switching and high temperature effects on LS and SG populations (Fig. 3–8). This difference may be attributed to the distinct genetic backgrounds of the LS and SG populations. Genetic background is important for the performance of microbiota, including symbionts in host insects (73). For example, the same *Cardinium* strain can confer various fitness manipulations in different geographical populations of *B. tabaci* that have heterogeneous genetic backgrounds (36). The effects of high temperatures on LS and SG populations were also disparate, likely due to their diverse genetic backgrounds (22). The performance of *Rickettsia* varies across different populations of *B. tabaci* genotypes (74). Additionally, reproductive manipulations of *Wolbachia* and *Spiroplasma* co-infection in *Tetranychus truncatus* were also related to the genetic backgrounds of host spider mites (35, 75). Another study also revealed a close co-evolutionary relationship between symbionts and their hosts (76). The close relationship between host genotype and microbiota should be considered in related research.

## Limitations of relative abundance data

High-throughput sequencing of marker genes, such as the 16S rRNA gene, has become a standard approach for profiling microbial communities. These methods typically provide information on the relative abundance of different taxa within a sample. However, relying solely on relative abundance data has several limitations (77, 78). Relative abundances do not account for changes in the total microbial load and can be influenced by technical biases such as differences in DNA extraction efficiency, PCR amplification, and 16S rRNA gene copy number variation among taxa. As a result, apparent changes in the proportion of one taxon may reflect changes in other community members rather than true shifts in absolute abundance. This compositional nature of sequencing data can lead to misinterpretation of microbial dynamics and ecological relationships (79, 80). In this experiment, full-length 16S rRNA gene sequencing and relative qPCR results were all relative abundance data, which may be limited. To overcome these limitations, absolute quantification methods, such as absolute qPCR or the use of internal standards in future studies, are recommended to provide a more accurate assessment of microbial population sizes.

In summary, we reported various effects of the host plants and high temperatures on the bacterial communities of LS and SG populations. This includes both *Cardinium*-infected and uninfected *B. tabaci* isofemale lines. We discovered that high temperature could significantly decrease microbiota diversity in the LS and SG populations. In contrast, host-plant switching and high temperature significantly changed the symbiont abundance and functions of the bacteria in whiteflies. However, these effects were also associated with *Cardinium*-infected status and host genetic backgrounds, suggesting that the composition of the bacterial community in *B. tabaci* MED is shaped by a combination of symbiont infection, host genetics, and host plant factors. The potential interactive or mixed effects among these variables highlight the complexity of the system. A multivariate analysis could be employed in future studies to disentangle and quantify the relative contributions and interactions of these factors. The next step is to explore which specific factors in the host plant influence the bacterial community and how *Cardinium* infection affects other bacteria (mainly symbionts) in *B. tabaci* MED.

### ACKNOWLEDGMENTS

This research was supported by the National Natural Science Foundation of China (32372534), the Shandong Provincial Natural Science Foundation (ZR2024QC128), the Taishan Scholar Foundation of Shandong Province, and the Qingdao Agricultural University High-level Talent Fund (663-1121025).

Conceptualization: C.D.; sample preparation: Q.P.H. and Y.M.Y.; methodology and data analysis: Y.K. and L.C.R.; qPCR experiment: Q.P.H.; writing and editing: Y.K., L.C.R., and C.D.; funding acquisition: C.D. and Y.K.

### AUTHOR AFFILIATIONS

[1]Shandong Engineering Research Center for Environment-friendly Agricultural Pest Management, College of Plant Health and Medicine, Qingdao Agricultural University, Qingdao, China

[2]Shandong Province Centre for Bioinvasions and Eco-security, College of Plant Health and Medicine, Qingdao Agricultural University, Qingdao, China

[3]Key Laboratory of Integrated Pest Management on Crops in East China, Ministry of Agriculture and Rural Affairs, Nanjing Agricultural University, Nanjing, Jiangsu, China

### AUTHOR ORCIDs

Kun Yang  http://orcid.org/0000-0001-9443-6494
Dong Chu  http://orcid.org/0000-0002-0648-9236

## FUNDING

| Funder | Grant(s) | Author(s) |
|---|---|---|
| National Nature Science Foundation of China | 32372534 | Dong Chu |
| Shandong Provincial Natural Science Foundation | ZR2024QC128 | Kun Yang |
| Taishan Scholar Foundation of Shandong Province | | Dong Chu |
| Qingdao Agricultural University High-level Talent Fund | 663-1121025 | Kun Yang |

## AUTHOR CONTRIBUTIONS

Kun Yang, Conceptualization, Data curation, Formal analysis, Funding acquisition, Methodology, Writing – original draft, Writing – review and editing | Cheng-Ran Li, Conceptualization, Data curation, Investigation, Methodology, Writing – original draft | Peng-Hao Qin, Data curation, Formal analysis, Methodology, Validation | Meng-Ying Yuan, Data curation, Formal analysis, Methodology, Writing – original draft | Dong Chu, Conceptualization, Data curation, Formal analysis, Funding acquisition, Methodology, Writing – original draft, Writing – review and editing

## DATA AVAILABILITY

The raw reads of 16S rRNA sequencing have been deposited in the NCBI Sequence Read Archive (SRA) database (BioProject ID: SUB14449026, https://submit.ncbi.nlm.nih.gov/subs/sra/SUB14449026/overview).

## ADDITIONAL FILES

The following material is available online.

### Supplemental Material

**Supplemental figures and table (Spectrum02240-24-s0001.docx).** Fig. S1 to S4 and Table S1.

### Open Peer Review

**PEER REVIEW HISTORY (review-history.pdf).** An accounting of the reviewer comments and feedback.

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
