## [Reviewer comments · Microbiology Spectrum]

Microbiology Spectrum

Conditional effects of *Cardinium* on microbiota in an invasive whitefly under different ecological factors

Kun Yang, Chengran Li, Penghao Qin, Mengying Yuan, and Dong Chu

Corresponding Author(s): Dong Chu, Qingdao Agricultural University

Review Timeline:

Submission Date:	September 6, 2024
Editorial Decision:	November 11, 2024
Revision Received:	November 27, 2024
Editorial Decision:	April 24, 2025
Revision Received:	May 26, 2025
Editorial Decision:	June 9, 2025
Revision Received:	June 17, 2025
Accepted:	July 7, 2025

Editor: Adam C.N. Wong

Reviewer(s): Disclosure of reviewer identity is with reference to reviewer comments included in decision letter(s). The following individuals involved in review of your submission have agreed to reveal their identity: Xiaoli Bing (Reviewer #2); Yu-Hao Huang (Reviewer #3)

Transaction Report:

DOI: <https://doi.org/10.1128/spectrum.02240-24>

Re: Spectrum02240-24 (**Conditional effects of *Cardinium* on microbiota in an invasive whitefly under different ecological factors**)

Dear Prof. Dong Chu:

Thank you for the privilege of reviewing your work. Below you will find my comments, instructions from the Spectrum editorial office, and the reviewer comments.

Sincerely,
Guido Favia
Editor
Microbiology Spectrum

Reviewer #1 (Comments for the Author):

The paper by Yang et al., concerns the effect of elevated temperatures and plant-switches on the microbiota of whiteflies. In general, the message of the paper is clear, but some revisions are warranted as described below.

In order of appearance:

Line 80: microbiome->microbiota

Line 95: it should say "On the other hand,". But when used, the expression should be used on conjunction with "On the one hand,".

Line 108: lines->line.

It seems that the first two subheadings in Material and methods do not correspond well to the text below the headings. Please adjust.

Under the heading Sample preparation for 16s rRNA gene analysis and qPCR it would help the reader to show the set-up in a schematic figure (potentially for supplementary data).

Lines 208-211: how do you differ between primary and secondary symbiont?

Lines 211-212: what does "One symbiont genus has one symbiont species" mean?

Line 213: Firicutes->Firmicutes

Line 216: were associated -> was associated

Not clear if lines 238-240 are two different headings that was supposed to be at different levels.

Line 396: curcial->crucial

Please check the use of the word "would" in the discussion.

Are the symbionts found in the host plants? Lines 430-432?

Dear Editor and Reviewer:

With the kind comments of reviewer, we improved the manuscript, with the detailed information in following text:

1. Q: Line 80: microbiome->microbiota

A: Corrected.

2. Line 95: it should say "On the other hand,". But when used, the expression should be used on conjunction with "On the one hand,".

A: Deleting "on the other side" in line 94.

3. Line 108: lines->line.

A: Corrected.

4. It seems that the first two subheadings in Material and methods do not correspond well to the text below the headings. Please adjust.

A: Corrected the first two subheadings in Material and methods.

5. Under the heading Sample preparation for 16s rRNA gene analysis and qPCR it would help the reader to show the set-up in a schematic figure (potentially for supplementary data).

A: Here we provide the schematic figure of previous section "16s rRNA gene analysis and qPCR", and we think it is not necessary to list the figure in supplemental data.

after rearing for 1 generation, 1-day-old female adults were selected for 16s rDNA sequencing

6. Line 208-211: how do you differ between primary and secondary symbiont?

A: Primary symbiont is regarded as the necessary bacteria for arthropods, as well as in whitefly. Based on many other research, *Portiera aleyrodidarum* is the only primary symbiont for whitefly, which is necessary for the growth

and nutrition for host whitefly. As for other symbionts, including *Cardinium hertigii*, *Rickettsia* sp. strain MEAM1 and *Hamiltonella defensa*, which symbionts are not necessary for whitefly, these bacteria may be vital for many aspects of host whitefly, while whitefly won't die without these secondary symbionts. As for primary symbiont *Portiera aleyrodidarum*, whitefly could not live without this bacterium.

7. Line 211-212, what does "One symbiont genus has one symbiont species" mean?

A: Detailed information added in manuscript lines 214-215.

8. Line 213: Firicutes->Firmicutes

A: Corrected.

9. Line 216: were associated -> was associated

A: Corrected.

10. Not clear if lines 238-240 are two different headings that was supposed to be at different levels.

A: Adding "symbionts' titers and diversity of" in line 241 and 342 to clarify the headings.

11. Line 396: curcial->crucial

A: Corrected.

12. Please check the use of the word "would" in the discussion.

A: The uses of "would" in Discussion are fully modified, throughout all discussion part.

13. Are the symbionts found in the host plants? Lines 430-432?

A: Based on PCR detection of plant leaves' DNA with symbionts' specific primers, no symbionts in host plants (including cotton and tomato) were discovered.

Re: Spectrum02240-24R1 (**Conditional effects of *Cardinium* on microbiota in an invasive whitefly under different ecological factors**)

Dear Prof. Dong Chu:

Thank you for submitting your work to Microbiology Spectrum. Your manuscript has now been reviewed by three experts, who expressed enthusiasm of the work but also indicated several caveats. My decision is to revise the manuscript to thoroughly address the reviewers' comments.

Revision Guidelines

Sincerely,
Adam C.N. Wong
Editor
Microbiology Spectrum

Reviewer #1 (Public repository details (Required)):

It says: "The raw reads of 16S rRNA sequencing have been deposited in the NCBI Sequence Read Archive (SRA) database (BioProject ID: PRJNA1111653)". But I cannot find it there.

Reviewer #1 (Comments for the Author):

1. As English is not the first language of the authors, I strongly recommend that the figure produced for the response to reviewer regarding Sample preparation for 16s rRNA gene analysis and qPCR to be added. Either as a figure in the manuscript or as supplementary data.

2. Regarding the difference between first and secondary symbiont, the authors write in lines 408-409 that: "These results showed that the secondary symbiont *Hamiltonella* was crucial for the growth and development of *B. tabaci* MED".

How does this align with their response:

"A: Primary symbiont is regarded as the necessary bacteria for arthropods, as well as in whitefly. Based on many other research, *Portiera aleyrodidarum* is the only primary symbiont for whitefly, which is necessary for the growth and nutrition for host whitefly. As for other symbionts, including *Cardinium hertigii*, *Rickettsia* sp. strain MEAM1 and *Hamiltonella defense*, which symbionts are not necessary for whitefly, these bacteria may be vital for many aspects of host whitefly, while whitefly won't die without these secondary symbionts. As for primary symbiont *Portiera aleyrodidarum*, whitefly could not live without this bacterium."

3. I cannot find the statement that the plants are not having symbionts.

Q: Are the symbionts found in the host plants? Lines 430-432?

A: Based on PCR detection of plant leaves' DNA with symbionts' specific primers, no symbionts in host plants (including cotton and tomato) were discovered. [I guess the authors mean tobacco and not tomato.]

Reviewer #2 (Comments for the Author):

This study investigates the impact of *Cardinium* infection on the microbiota in *Bemisia tabaci* MED populations with different genetic backgrounds under varying ecological factors, particularly high temperatures and host-plant changes. Using full-length 16S rRNA gene sequencing and qPCR experiments, they analyzed two geographical populations of *B. tabaci* MED. Results show that high temperature treatment and host-plant switching differentially affect endosymbiont titer, microbiota diversity, and function in these populations, highlighting the role of genetic background. Notably, an increase in *Cardinium* titer significantly alters microbiota function in infected lines compared to uninfected ones, while decreases or stable titers do not, suggesting a close association between *Cardinium* titer and microbiota function. These findings enhance the understanding of the complex interplay among endosymbionts, microbiota, and host insects, and emphasize the importance of genetic background in shaping microbiota responses to environmental stressors.

Compared to the original version, the revised paper has been significantly improved. However, there may be some minor details that need to be corrected.

Table 1 should provide more detailed information and is recommended to be moved to Table S1 as supplementary material.

The titles on pages 251 and 274 are repeated from page 250. Please remove them.

In the merged PDF file, figure legends are missing, and the related information is absent. As a result, it is unclear what each figure represents and how they were created.

Figure 2 presents a phylogenetic tree, but it requires clearer annotations for several pieces of information.

Have the authors confirmed that the abundance of *Rickettsia* is indeed higher than that of the primary symbiont *Portiera* in whiteflies through absolute quantitative PCR? This is a crucial finding that needs verification. The authors report that the primary bacterium *Portiera* accounts for less than 30% of most populations of the tobacco whiteflies, while *Rickettsia* makes up a significantly higher percentage, averaging over 50%. This finding is inconsistent with the typical symbiotic bacterial composition observed in whiteflies, and it is recommended that the authors discuss this discrepancy.

Is Figure 3 based on absolute quantitative PCR, or was it derived from the reads obtained through microbiota sequencing?

Reviewer #3 (Comments for the Author):

The article investigates the effects of *Cardinium* infection, population, temperature, and host plant on the titer of four main whitefly endosymbionts (*Portiera*, *Hamiltonella*, *Cardinium*, and *Rickettsia*) through qPCR experiments, as well as the diversity and functional abundance of the microbiota using full-length 16S rRNA gene sequencing in *Bemisia tabaci* MED. The authors have conducted extensive work and describe the combined effects of these factors on whitefly endosymbionts and the overall microbiota. Overall, the methodology is sound, and the main conclusions are clearly presented.

Below are the detailed comments or suggestions that could improve clarity.

Line 27-30: It would be beneficial to mention ecological factors such as high-temperature treatment and host-plant switching in

the research contents.

Line 43: The italicized MED differs from its formatting in the abstract.

Line 81-85: These two sentences seem redundant.

Line 86: "the diversities of host *Sogatella furcifera*": Is this correct, or does it refer to the diversity of bacteria?

Line 89-91: This description lacks specific examples and references.

Line 92-93: "symbiotic bacteria and the microbiota": I think that similar expression should be improved in the whole manuscript. Symbiotic bacteria are members of the microbiota, so changes in symbiotic bacteria naturally alter the microbiota. While the meaning is understandable, the expression could be clearer and more accurate.

Line 95-100: These two sentences are somewhat redundant and could be refined for clarity.

Line 111: The full names of the two populations should be provided here. Additionally, it would be helpful to know if these populations exhibit significant genetic differentiation. If this has been reported, it should be mentioned in the introduction, as this would enhance the persuasiveness of the conclusions regarding genetic effects.

Line 156-157: Is one day sufficient for the development from egg to adult? Also, the methods for identifying these female adults should be described.

Line 204: The references for these three tools should be cited. Furthermore, the methods for phylogenetic and functional analyses are lacking.

Line 217: Why is there more than one value for the average length? Does this represent a range?

Line 221-223: The sentence appears incomplete.

Line 240: How are endosymbionts and symbionts mentioned elsewhere defined? In other words, which bacteria are included in this category? It would be beneficial to emphasize the definitions of endosymbionts/symbionts in the introduction, as well as distinguish them from other bacteria, particularly when involving this conception in the conclusion.

Line 257: Is this referring to Fig. 3?

Line 391: Fig. 3 seems to contradict the statement that high temperature reduces the abundance of symbionts.

Line 393: Italic "infection status and".

Line 407: *Portiera* should be italicized.

Line 477: Capitalized Vitellogenin.

Line 519-520: I think that there may be something wrong in this sentence because the evolutionary relationship between symbiotic bacteria and insect hosts is not close.

Line 530: The mixed or interactive effects should be summarized and described, or a multivariate analysis could be considered.

Line 811-817: The methods mentioned here are absent from the methods section.

Fig. 2: Some trends in the relative abundance of the four main bacteria appear inconsistent with the qPCR results, such as *Cardinium* under different host plant treatments for the SGC+ population. Please verify the results. If both are correct, it is likely that the discrepancy arises from differences in detection sensitivity and parameters (relative abundance versus relative amounts). However, the difference should be discussed to prevent misinterpretation by readers.

The article investigates the effects of *Cardinium* infection, population, temperature, and host plant on the titer of four main whitefly endosymbionts (*Portiera*, *Hamiltonella*, *Cardinium*, and *Rickettsia*) through qPCR experiments, as well as the diversity and functional abundance of the microbiota using full-length 16S rRNA gene sequencing in *Bemisia tabaci* MED. The authors have conducted extensive work and describe the combined effects of these factors on whitefly endosymbionts and the overall microbiota. Overall, the methodology is sound, and the main conclusions are clearly presented.

Below are the detailed comments or suggestions that could improve clarity.

Line 27-30: It would be beneficial to mention ecological factors such as high-temperature treatment and host-plant switching in the research contents.

Line 43: The italicized MED differs from its formatting in the abstract.

Line 81-85: These two sentences seem redundant.

Line 86: “the diversities of host *Sogatella furcifera*”: Is this correct, or does it refer to

the diversity of bacteria?

Line 89-91: This description lacks specific examples and references.

Line 92-93: "symbiotic bacteria and the microbiota": I think that similar expression should be improved in the whole manuscript. Symbiotic bacteria are members of the microbiota, so changes in symbiotic bacteria naturally alter the microbiota. While the meaning is understandable, the expression could be clearer and more accurate.

Line 95-100: These two sentences are somewhat redundant and could be refined for clarity.

Line 111: The full names of the two populations should be provided here. Additionally, it would be helpful to know if these populations exhibit significant genetic differentiation. If this has been reported, it should be mentioned in the introduction, as this would enhance the persuasiveness of the conclusions regarding genetic effects.

Line 156-157: Is one day sufficient for the development from egg to adult? Also, the methods for identifying these female adults should be described.

Line 204: The references for these three tools should be cited. Furthermore, the methods for phylogenetic and functional analyses are lacking.

Line 217: Why is there more than one value for the average length? Does this represent a range?

Line 221-223: The sentence appears incomplete.

Line 240: How are endosymbionts and symbionts mentioned elsewhere defined? In other words, which bacteria are included in this category? It would be beneficial to emphasize the definitions of endosymbionts/symbionts in the introduction, as well as distinguish them from other bacteria, particularly when involving this conception in the conclusion.

Line 257: Is this referring to Fig. 3?

Line 391: Fig. 3 seems to contradict the statement that high temperature reduces the abundance of symbionts.

Line 393: Italic "infection status and".

Line 407: Portiera should be italicized.

Line 477: Capitalized Vitellogenin.

Line 519-520: I think that there may be something wrong in this sentence because the evolutionary relationship between symbiotic bacteria and insect hosts is not close.

Line 530: The mixed or interactive effects should be summarized and described, or a multivariate analysis could be considered.

Line 811-817: The methods mentioned here are absent from the methods section.

Fig. 2: Some trends in the relative abundance of the four main bacteria appear inconsistent with the qPCR results, such as *Cardinium* under different host plant treatments for the SGC+ population. Please verify the results. If both are correct, it is likely that the discrepancy arises from differences in detection sensitivity and

parameters (relative abundance versus relative amounts). However, the difference should be discussed to prevent misinterpretation by readers.

Dear Editor and Reviewer:

With the kind comments of reviewer, we improved the manuscript, with the detailed information in following text:

Q: "The raw reads of 16S rRNA sequencing have been deposited in the NCBI Sequence Read Archive (SRA) database (BioProject ID: PRJNA1111653)". But I cannot find it there.

A: Thanks for your question, and the submit number is "SUB14449026" in NCBI database, we corrected related contents in manuscript.

Responses to Reviewer #2:

- Q: As English is not the first language of the authors, I strongly recommend that the figure produced for the response to reviewer regarding Sample preparation for 16s rRNA gene analysis and qPCR to be added. Either as a figure in the manuscript or as supplementary data.**

A: Thanks for your kind comments, and we provide the following figure as

Fig. S1 to describe 16s rRNA gene analysis and qPCR experiments.

2. **Q: Regarding the difference between first and secondary symbiont, the authors write in lines 408-409 that: "These results showed that the secondary symbiont *Hamiltonella* was crucial for the growth and development of *B. tabaci* MED".**

How does this align with their response:

"A: Primary symbiont is regarded as the necessary bacteria for arthropods, as well as in whitefly. Based on many other research, *Portiera aleyrodidarum* is the only primary symbiont for whitefly, which is necessary for the growth and nutrition for host whitefly. As for other symbionts, including *Cardinium hertigii*, *Rickettsia* sp. strain MEAM1 and *Hamiltonella* defense, which symbionts are not necessary for whitefly, these bacteria may be vital for many aspects of host whitefly, while whitefly won't die without these secondary symbionts. As for primary symbiont *Portiera aleyrodidarum*, whitefly could not live without this bacterium."?

A: There is no contradiction between these statements, but rather a difference in the degree of necessity for the host. The primary symbiont, *Portiera aleyrodidarum*, is absolutely essential for the survival of *B. tabaci*; without it, the insect cannot live. In contrast, secondary symbionts such as *Hamiltonella* are not strictly required for survival—*B. tabaci* can live without them. However, our results and other studies have shown that secondary symbionts like *Hamiltonella* can play a crucial role in optimizing the growth and development of the host. In other words, while *Hamiltonella* is not essential for basic survival, it is important for the host's optimal fitness, development, or adaptation to certain conditions. Therefore, the term "crucial" in our manuscript refers to the significant positive impact of *Hamiltonella* on host development, not to an absolute requirement for survival as is the case for the primary symbiont.

3. **I cannot find the statement that the plants are not having symbionts.**

Q: Are the symbionts found in the host plants? Lines 430-432?

A: Based on PCR detection of plant leaves' DNA with symbionts' specific primers, no symbionts in host plants (including cotton and tomato) were discovered. [I guess the authors mean tobacco and not tomato.]

A: Thank you for your comment and for pointing out the need for clarification. In our study, we performed PCR detection using symbiont-specific primers on DNA extracted from the leaves of host plants (including cotton and tobacco). The results showed that no symbionts were detected in any of the host plant samples. This indicates that the symbionts investigated in our study are not present in the host plants themselves but are restricted to the whitefly hosts. We have revised the manuscript to clarify this point and corrected the plant

species to “tobacco” instead of “tomato” where appropriate in lines 270 to 273.

Responses to Reviewer #2:

1. **Q: Table 1 should provide more detailed information and is recommended to be moved to Table S1 as supplementary material.**

A: Thank you for your suggestion. We agree that Table 1 contains detailed sequencing information that may be more appropriate as supplementary material. Accordingly, we have moved Table 1 to the supplementary section and renamed it as Table S1. In the main text, we have updated the reference to this table in line 235.

2. **Q: The titles on pages 251 and 274 are repeated from page 250. Please remove them.**

A: Repeated titles were removed.

3. **In the merged PDF file, figure legends are missing, and the related information is absent. As a result, it is unclear what each figure represents and how they were created.**

A: Thank you for bringing this to our attention. We apologize for the omission of figure legends in the merged PDF file. We have carefully checked the merged PDF to ensure that all figure legends and related information are now present and correctly formatted.

4. **Figure 2 presents a phylogenetic tree, but it requires clearer annotations for several pieces of information.**

A: Fig. 1 is the phylogenetic tree, and detailed information added in legend: “Circular phylogenetic tree showing the evolutionary relationships among bacterial genera isolated from *Bemisia tabaci* MED individuals from the LS and SG populations. The tree was constructed using QIIME (v1.9.0) based on 16S rRNA gene sequencing data. Each tip of the tree represents a bacterial genus or species, with branch colors corresponding to different bacterial phyla as indicated in the legend. Closely related taxa are clustered together, illustrating their shared evolutionary history. Branch lengths reflect evolutionary distances between taxa.”

5. **Have the authors confirmed that the abundance of *Rickettsia* is indeed higher than that of the primary symbiont *Portiera* in whiteflies through absolute quantitative PCR? This is a crucial finding that needs**

verification. The authors report that the primary bacterium *Portiera* accounts for less than 30% of most populations of the tobacco whiteflies, while *Rickettsia* makes up a significantly higher percentage, averaging over 50%. This finding is inconsistent with the typical symbiotic bacterial composition observed in whiteflies, and it is recommended that the authors discuss this discrepancy.

A: We acknowledge the importance of validating high-throughput sequencing results with absolute quantitative PCR (qPCR). In our study, the reported abundances of *Rickettsia* and *Portiera* were based on 16S rRNA gene amplicon sequencing data (Fig. 2) and relative qPCR experiment (Fig. 3 and 6), while no absolute quantification using qPCR in this study, and we recognize that sequencing-based relative abundances can be influenced by factors such as primer bias and differences in 16S rRNA gene copy number among bacteria. However, full-length 16s rRNA gene sequencing and relative qPCR experiment both confirmed that *Rickettsia* abundance was higher than that of *Portiera* (Fig. 2, 3 and 6), and we have added a statement in the Discussion section acknowledging the limitation of using only relative abundance data in lines 552 to 568.

6. Is Figure 3 based on absolute quantitative PCR, or was it derived from the reads obtained through microbiota sequencing?

A: Figure 3 based on relative qPCR results described in Materials and Methods section in lines 174 to 191.

Responses to Reviewer #3:

1. Line 27-30: It would be beneficial to mention ecological factors such as high-temperature treatment and host-plant switching in the research contents.

A: Detailed information added in lines 29 to 31.

2. Line 43: The italicized MED differs from its formatting in the abstract.

A: Corrected.

3. Line 81-85: These two sentences seem redundant.

A: Deleting previous sentences in 81-85.

- 4. Line 86: "the diversities of host *Sogatella furcifera*": Is this correct, or does it refer to the diversity of bacteria?**

A: Sorry this is our mistake, in manuscript we added "microbiota in" after "the diversities".

- 5. Line 89-91: This description lacks specific examples and references.**

A: Specific examples and references added in lines 90-91.

- 6. Line 92-93: "symbiotic bacteria and the microbiota": I think that similar expression should be improved in the whole manuscript. Symbiotic bacteria are members of the microbiota, so changes in symbiotic bacteria naturally alter the microbiota. While the meaning is understandable, the expression could be clearer and more accurate.**

A: Thanks for your comments, in line 93 the description changed to "symbiotic bacteria and other bacterial microbiota", and the similar descriptions revised throughout all manuscript.

- 7. Line 95-100: These two sentences are somewhat redundant and could be refined for clarity.**

A: Contents refined in manuscript lines 96-101.

- 8. Line 111: The full names of the two populations should be provided here. Additionally, it would be helpful to know if these populations exhibit significant genetic differentiation. If this has been reported, it should be mentioned in the introduction, as this would enhance the persuasiveness of the conclusions regarding genetic effects.**

A: Full names of the 2 populations added in manuscript, and the genetic backgrounds are totally different between LS and SD whitefly populations as revealed in our previous study (Li et al., 2023), and the related contents added in manuscript line 128-129.

- 9. Line 156-157: Is one day sufficient for the development from egg to adult? Also, the methods for identifying these female adults should be described.**

A: Thanks for your comments, and 1-day-old female adults mean the whitefly after emergence within 24 hours.

- 10. Line 204: The references for these three tools should be cited. Furthermore, the methods for phylogenetic and functional analyses are lacking.**

A: Related references and contents added in manuscript lines 219-223, as well as in references section.

11. Line 217: Why is there more than one value for the average length? Does this represent a range?

A: The 16S rRNA gene is generally about 1,500 base pairs (bp) long, but its exact length can vary slightly between different bacterial species. This variation is due to insertions or deletions (indels) in certain regions of the gene, especially in the hypervariable regions (V1–V9).

12. Line 221-223: The sentence appears incomplete.

A: The sentence is polished in manuscript lines 239-243.

13. Line 240: How are endosymbionts and symbionts mentioned elsewhere defined? In other words, which bacteria are included in this category? It would be beneficial to emphasize the definitions of endosymbionts/symbionts in the introduction, as well as distinguish them from other bacteria, particularly when involving this conception in the conclusion.

A: Symbionts refer to bacteria that live in close association with the whitefly host. Endosymbionts are defined as bacteria that reside within the cells or specialized tissues (such as bacteriocytes) of the host and are typically vertically transmitted from parent to offspring. In this article, as well as for whitefly, all symbionts including *Portiera*, *Cardinium*, *Hamiltonella* and *Rickettsia* are all endosymbionts. Based on these fundament, we revised all “endosymbionts” to “symbionts” in this article.

14. Line 257: Is this referring to Fig. 3?

A: Thanks for your comments, this is referred to Fig. 3, and the mistake corrected in manuscript.

15. Line 391: Fig. 3 seems to contradict the statement that high temperature reduces the abundance of symbionts.

A: Thanks for your comments, the related contents of “reduced” replaced with “significantly altered” in manuscript.

16. Line 393: Italic "infection status and".

A: Corrected.

17. Line 407: *Portiera* should be italicized.

A: Corrected.

18. Line 477: Capitalized Vitellogenin.

A: Corrected.

19. Line 519-520: I think that there may be something wrong in this sentence because the evolutionary relationship between symbiotic bacteria and insect hosts is not close.

A: Thanks for your comments, and the sentence as written is misleading. Symbiotic bacteria and their insect hosts are not closely related in a phylogenetic sense, while there is a close co-evolutionary relationship or long-term association between symbionts and hosts, not a close evolutionary relationship. Here are the revised sentences in manuscript: “Another study also revealed a close co-evolutionary relationship between symbionts and their hosts (Fisher et al., 2017).”

20. Line 530: The mixed or interactive effects should be summarized and described, or a multivariate analysis could be considered.

A: Thanks for your kind comments, and related contents revised in manuscript as follows: “However, these effects were also associated with *Cardinium*-infected status and host genetic backgrounds, suggesting that the composition of the bacterial community in *B. tabaci* MED is shaped by a combination of symbiont infection, host genetics, and host plant factors. The potential interactive or mixed effects among these variables highlight the complexity of the system. A multivariate analysis could be employed in future studies to disentangle and quantify the relative contributions and interactions of these factors. The next step is to explore which specific factors in the host plant influence the bacterial community and how *Cardinium* infection affects other bacteria (mainly symbionts) in *B. tabaci* MED.”

21. Line 811-817: The methods mentioned here are absent from the methods section.

A: Thanks for your kind comments, and CCS (Circular Consensus Sequencing) sequences are outputs of results described in manuscript lines 192-213.

22. Fig. 2: Some trends in the relative abundance of the four main bacteria appear inconsistent with the qPCR results, such as *Cardinium* under different host plant treatments for the SGC+ population. Please verify the results. If both are correct, it is likely that the discrepancy arises from differences in detection sensitivity and parameters (relative abundance

versus relative amounts). However, the difference should be discussed to prevent misinterpretation by readers.

A: Thanks for your kind comments, and we believe relative qPCR experiment is more accurate. Relative contents added in manuscript Introduction section from lines 115-122: “16S rRNA gene sequencing is widely used for profiling bacterial communities, while its accuracy can be affected by primer bias (Some bacterial taxa may be under- or over-represented due to mismatches with universal primers), PCR amplification bias, resolution (closely related species may not be distinguished due to conserved regions in the 16S rRNA gene) (Janda, J. M., & Abbott, 2007). Relative qPCR is a sensitive and specific method for quantifying the abundance of target DNA sequences, such as specific bacterial taxa. Its accuracy depends on highly primer specificity, accurate PCR efficiency and results are often normalized to a reference gene (Smith and Osborn,2009).”

Re: Spectrum02240-24R2 (**Conditional effects of *Cardinium* on microbiota in an invasive whitefly under different ecological factors**)

Dear Prof. Dong Chu:

Thank you for the privilege of reviewing your work. Below you will find my comments, instructions from the Spectrum editorial office, and the reviewer comments.

All three reviewers agreed that the manuscript has been significantly improved in the revised version. There remain some minor issues that need addressed.

Revision Guidelines

Sincerely,
Adam C.N. Wong
Editor
Microbiology Spectrum

Reviewer #1 (Public repository details (Required)):

The new submission number SUB14449026 is not possible to find at NCBI.

Reviewer #1 (Comments for the Author):

Thank you for adding figure S1. I suggest adding the text from the material and methods to make it even more clear what is shown. There are some signs that I don't understand

Top left there is a plus sign with an arrow, what does it mean?

Top right there is the insect a minus sign and maybe a snow flake or a star.

In lines 408-409, change the word from crucial to something less strong.

Reviewer #2 (Comments for the Author):

Compared to the original version, the revised paper has been significantly improved. I have only some minor details that need to be corrected.

The phylogenetic tree in Figure 1 still lacks some important elements. There are neither bootstrap values nor a scale bar indicating branch length.

I think instead of discussing the limitations of relative abundance here, the author could conduct an additional experiment to calculate the absolute copy number of the samples used for qPCR. All that's needed is to create a standard curve.

In line 93, the author changed it to "symbiotic bacteria and other bacterial microbiota", but this expression is also inappropriate. "Bacterial microbiota" is not accurate enough.

Lines 99 - 106, this content actually doesn't need to be placed in the Introduction as it will affect the presentation of the paper. If the author insists on emphasizing these methods, they can be put in the Materials and Methods section. In fact, these methods are quite traditional and not very complex, so there's no need for an introduction.

Why does "Human disease" appear in the metabolic pathway enrichment information in Figure 5 and 8? This indicates that the corresponding KEGG annotation is abnormal. There can't be any pathways related to these in the symbiotic bacteria of the whitefly. This part of the results is meaningless, and it is recommended that the author remove it.

Reviewer #3 (Comments for the Author):

The authors have made significant improvements in response to the previous round of reviews. Most of my prior comments have been satisfactorily addressed, and the manuscript has been strengthened as a result. However, regarding the newly added content, a few descriptions require further improvements to enhance grammar, clarity, and readability.

Line 99-106: The paragraph of method introduction feels somewhat abrupt. I recommend adding transitional sentences (e.g., "These methods are well-suited to explore the research question") or integrating this content with the limitations of the methods in Discussion section.

Line 109: The sentence should be revised as follows: "with genetic backgrounds distinct from ..."

Line 199-203: I strongly suggest providing a more detailed description of the methods or tools used for CCS sequence management (currently listed under Table S1) and phylogenetic analysis (e.g., MAFFT + FastTree/IQ-TREE?).

Line 218: "1425 bp and 1466 bp": To my understanding, the numbers here should indicate a range. If this is the case, the sentence would be clearer if revised to: "The average length of sequenced CCS in each sample ranged from 1425 bp to 1466 bp."

Reviewer #1 (Public repository details (Required)):

1. Q: The new submission number SUB14449026 is not possible to find at NCBI.

A: Thanks for your comments, and we provide a link about this submission in manuscript as <https://submit.ncbi.nlm.nih.gov/subs/sra/SUB14449026/overview>, the detailed information of this submission as follow figure:

All official websites of the United States government. Here's how you know ✓

 **National Library of Medicine**
National Center for Biotechnology Information

Submission Portal

Sequence Read Archive (SRA) submission: SUB14449026

16s rRNA gene sequencing data of Bemisia tabaci under different temperatures and host plants, May 14 '24

- ✓ **BioProject: Processed**
PRJNA1111653 : 16s rRNA gene sequencing data of Bemisia tabaci under different temperatures and host plants
- ✓ **BioSample: Processed**
Successfully loaded
(48 objects)
Download attributes file with BioSample accessions
- ✓ **SRA: Processed**
(48 objects)
Download metadata file with SRA accessions
View and manage my SRA submission data

Manage data

Summary

This Sequence Read Archive (SRA) submission will be released on **2025-06-11** or upon publication, whichever is first.

Note: Release of BioProject or BioSample is also triggered by the release of linked data.

BioSample attributes file MIMS.me.host-associated.xlsx (24.1 KB)

Metadata file SRA_metadat.xlsx (56.5 KB)

Sample name	Files	Status
Bacteria_A1_raw.fq	Bacteria_A1_raw.fq (fastq)	Processed
Bacteria_A2_raw.fq	Bacteria_A2_raw.fq (fastq)	Processed
Bacteria_A3_raw.fq	Bacteria_A3_raw.fq (fastq)	Processed
Bacteria_A4_raw.fq	Bacteria_A4_raw.fq (fastq)	Processed
Bacteria_B1_raw.fq	Bacteria_B1_raw.fq (fastq)	Processed
Bacteria_B2_raw.fq	Bacteria_B2_raw.fq (fastq)	Processed
Bacteria_B3_raw.fq	Bacteria_B3_raw.fq (fastq)	Processed
Bacteria_B4_raw.fq	Bacteria_B4_raw.fq (fastq)	Processed
Bacteria_C1_raw.fq	Bacteria_C1_raw.fq (fastq)	Processed
Bacteria_C2_raw.fq	Bacteria_C2_raw.fq (fastq)	Processed

More (38 rows)

2. Q: Thank you for adding figure S1. I suggest adding the text from the material and methods to make it even more clear what is shown. There are some signs that I don't

understand: Top left there is a plus sign with an arrow, what does it mean? Top right there is the insect a minus sign and maybe a snow flake or a star.

A: Thanks for your kind comments, and in previous study we would like to use “-” for Cardinium uninfected population, while “+” for Cardinium infected population, and in present Fig. S1 all signs deleted to make figure clearer.

3. Q: In lines 408-409, change the word from crucial to something less strong.

A: Changed “crucial” to “important”.

Reviewer #2 (Comments for the Author):

1. Q: The phylogenetic tree in Figure 1 still lacks some important elements. There are neither bootstrap values nor a scale bar indicating branch length.

A: Figure 1 remade with detailed information.

2. Q: I think instead of discussing the limitations of relative abundance here, the author could conduct an additional experiment to calculate the absolute copy number of the samples used for qPCR. All that's needed is to create a standard curve.

A: Thank you for your valuable feedback regarding the limitations of relative abundance data in our study. While we understand the importance of absolute quantification, we believe that discussing the limitations in our current manuscript is more appropriate for several reasons: 1) Study Focus: our main objective was to examine the relative changes in bacterial communities and symbiont titers under different ecological conditions. The relative abundance data provides sufficient insights into these changes and their biological significance. 2) Technical Considerations: while absolute quantification would be ideal, it requires careful calibration and validation for each target gene across different sample types. Given our focus on relative comparisons between treatments, the relative abundance data remains robust for our research questions. 3) Data interpretation: The limitations we discuss highlight important considerations for interpreting our results, which is valuable for readers and future researchers. This discussion adds depth to our analysis without necessitating additional experiments.

3. Q: In line 93, the author changed it to "symbiotic bacteria and other bacterial microbiota", but this expression is also inappropriate. "Bacterial microbiota" is not accurate enough.

A: Thank you for your valuable feedback regarding the terminology. In this version we use “bacterial taxa” instead of the ambiguous “bacterial microbiota”.

4. Q: Lines 99 - 106, this content actually doesn't need to be placed in the Introduction

as it will affect the presentation of the paper. If the author insists on emphasizing these methods, they can be put in the Materials and Methods section. In fact, these methods are quite traditional and not very complex, so there's no need for an introduction.

A: Thanks for your comments, and this paragraph added with previous reviewer's comments, and we moved this to Materials and Methods section in this version.

5. Q: Why does "Human disease" appear in the metabolic pathway enrichment information in Figure 5 and 8? This indicates that the corresponding KEGG annotation is abnormal. There can't be any pathways related to these in the symbiotic bacteria of the whitefly. This part of the results is meaningless, and it is recommended that the author remove it.

A: Deleting all data about human disease in these figures.

Reviewer #3 (Comments for the Author):

1. Q: Line 99-106: The paragraph of method introduction feels somewhat abrupt. I recommend adding transitional sentences (e.g., "These methods are well-suited to explore the research question") or integrating this content with the limitations of the methods in Discussion section.

A: Thanks for your advice, and based on your and other reviewer's comments, this paragraph moved to Materials and Methods section.

2. Q: Line 109: The sentence should be revised as follows: "with genetic backgrounds distinct from ..."

A: Corrected.

3. Q: Line 199-203: I strongly suggest providing a more detailed description of the methods or tools used for CCS sequence management (currently listed under Table S1) and phylogenetic analysis (e.g., MAFFT + FastTree/IQ-TREE?).

A: Detailed information about CCS and phylogenetic analysis added in Materials and Methods section.

4. Q: Line 218: "1425 bp and 1466 bp": To my understanding, the numbers here should indicate a range. If this is the case, the sentence would be clearer if revised to: "The average length of sequenced CCS in each sample ranged from 1425 bp to 1466 bp."

A: Corrected.

Re: Spectrum02240-24R3 (**Conditional effects of *Cardinium* on microbiota in an invasive whitefly under different ecological factors**)

Dear Prof. Dong Chu:

Your manuscript has been accepted, and I am forwarding it to the ASM production staff for publication. Your paper will first be checked to make sure all elements meet the technical requirements. ASM staff will contact you if anything needs to be revised before copyediting and production can begin. Otherwise, you will be notified when your proofs are ready to be viewed.

Sincerely,
Adam C.N. Wong
Editor
Microbiology Spectrum